# Repo-Man/PP1 regulates heterochromatin formation in interphase

Inês J. de Castro[1], James Budzak[1,†], Maria L. Di Giacinto[1], Lorena Ligammari[1], Ezgi Gokhan[1], Christos Spanos[2], Daniela Moralli[3], Christine Richardson[4], Jose I. de las Heras[2], Silvia Salatino[3], Eric C. Schirmer[2], Katharine S. Ullman[5], Wendy A. Bickmore[6], Catherine Green[3], Juri Rappsilber[2,7], Sarah Lamble[3], Martin W. Goldberg[4], Veronica Vinciotti[8] & Paola Vagnarelli[1]

Repo-Man is a protein phosphatase 1 (PP1) targeting subunit that regulates mitotic progression and chromatin remodelling. After mitosis, Repo-Man/PP1 remains associated with chromatin but its function in interphase is not known. Here we show that Repo-Man, via Nup153, is enriched on condensed chromatin at the nuclear periphery and at the edge of the nucleopore basket. Repo-Man/PP1 regulates the formation of heterochromatin, dephosphorylates H3S28 and it is necessary and sufficient for heterochromatin protein 1 binding and H3K27me3 recruitment. Using a novel proteogenomic approach, we show that Repo-Man is enriched at subtelomeric regions together with H2AZ and H3.3 and that depletion of Repo-Man alters the peripheral localization of a subset of these regions and alleviates repression of some polycomb telomeric genes. This study shows a role for a mitotic phosphatase in the regulation of the epigenetic landscape and gene expression in interphase.

[1] College of Health and Life Science, Research Institute for Environment Health and Society, Brunel University London, London UB8 3PH, UK. [2] Wellcome Trust Centre for Cell Biology, Edinburgh EH9 3BF, UK. [3] Wellcome Trust Centre for Human Genetics, University of Oxford, Oxford OX3 7BN, UK. [4] School of Biological and Medical Science, Durham University, Durham DH1 3LE, UK. [5] Huntsman Cancer Institute, University of Utah, Salt Lake City, Utah 84112, USA. [6] MRC Human Genetics Unit, Institute of Genetics and Molecular Medicine, University of Edinburgh, Edinburgh EH4 2XU, UK. [7] Technische Universitat Berlin, 13355 Berlin, Germany. [8] College of Engineering, Design and Technology, Research Institute for Environment Health and Society, Brunel University London, London UB8 3PH, UK. † Present address: Division of Cell and Molecular Biology, Department of Life Sciences, Imperial College London, London SW7 2AZ, UK. Correspondence and requests for materials should be addressed to P.V. (email: Paola.Vagnarelli@brunel.ac.uk).

The formation of the new G1 nucleus, after cells undergo mitosis, requires major re-organization and tight regulation of chromatin structure that together with nuclear envelope reformation provide the new cells with a nuclear environment containing essential cues for gene expression regulation[1].

Chromatin is mainly in a repressive state at the nuclear envelope, with the exception of regions around the nuclear pores (reviewed in Kind and van Steensel[2]). Peripheral chromatin is largely enriched in repressive histone modifications and heterochromatin protein 1 (HP1) that is anchored via its interaction with lamin B receptor[3]. Methylation of H3K9, thought to trigger association of chromatin to the lamina, and the polycomb-mediated H3K27me2/3 are particularly enriched at the nuclear periphery and at the edge of lamina-associated domains (reviewed in Bickmore *et al.*[4]). HP1 binding to H3K9me3 is enhanced in the presence of H3K27me3 and is blocked by phosphorylation of the adjacent H3S10 (refs 5–7), suggesting that a fine balance between these mechanisms culminates in a specific chromatin landscape and that phospho-methyl switches need to be tightly controlled during mitosis and in interphase.

Despite the emerging and recognized importance of protein phosphatases at M/G1 transition, very little is known about the details of how this class of enzymes regulates chromatin modifications and which phosphatases are essential for the reorganization of specific chromatin domains. Repo-Man (CDCA2) is a PP1 (protein phosphatase 1) targeting subunit[8] that, during mitotic exit, is essential for chromatin remodelling and nuclear envelope reformation[9] while in interphase is involved in DNA repair[10]. The Repo-Man/PP1 complex is targeted to chromatin at anaphase, where it de-phosphorylates Histone H3 (T3 and S10)[9,11], countering the mitotic kinases Haspin and Aurora B, respectively. These phospho-switches are essential for the removal of the chromosome passenger complex from the mitotic chromosomes (via H3T3)[12] thus allowing normal mitotic progression, and for the re-association of HP1 to H3K9me3 after mitosis (H3S10)[5]. Targeting of Repo-Man to chromatin is achieved via dephosphorylation of a chromatin-binding domain at the C terminus while the N terminus domain harbours the nuclear periphery targeting module and the binding site for Importin β (ref. 9). Once targeted to the chromatin in anaphase, the complex has a low turnover and PP1 is stably associated with Repo-Man[13]; therefore, from anaphase until the following mitosis, the complex could potentially act on chromatin locally and contribute to the maintenance of a specific chromatin landscape. However, the docking sites for Repo-Man on the chromosomes and the overall importance of the complex in interphase chromatin organization and maintenance are not known.

This study reveals that, in interphase, a fraction of Repo-Man associates with the heterochromatin beneath the inner lamina and adjacent to the nuclear pore complex (NPC). Repo-Man/PP1 is necessary for the organization of HP1 foci in interphase and sufficient to trigger a local enrichment of heterochromatin markers. We also provide evidence that Repo-Man contributes to the dephosphorylation of H3S28 with the potential to represent the counteracting phosphatase for the mitotic and stress kinases in interphase. Using an antibody-free technique that allows the investigation of protein–chromatin interactions, we show that Repo-Man associates with chromatin by binding directly to the modified lysine 27 on the H3 tail. Subtelomeric regions are particularly enriched for Repo-Man-binding sites where the complex contributes to generate a chromatin environment that is important for the peripheral localization and transcription regulation of a subset of telomeric regions.

Collectively, our data shows that Repo-Man/PP1 regulates the histone code and chromatin structure at least across a panel of target regions.

## Results

**Repo-Man associates with the nuclear envelope via Nup153.** Repo-Man associates with the chromosomes in anaphase and contributes to the assembly of nuclear envelope (NE) proteins for the formation of the new G1 nucleus. Published mass spectrometry analyses identified interactions of Repo-Man with several nuclear envelope proteins, namely, Importin β and Nup153 together with histone proteins[9,14] (Fig. 1a). While the interaction between Repo-Man and Importin β is direct, the link with the Nucleoporin Nup153 and its biological relevance is still unclear.

Repo-Man and Nup153 show some co-localization at the nuclear periphery (at deconvolution-microscopy resolution) (Fig. 1b). To understand this interaction at higher resolution, we first conducted proximity ligation assays (PLA) with antibodies against endogenous Repo-Man and Nup153; although both proteins are present within the entire nuclear space (Fig. 1c, panels a–f), PLA reveals that they interact at the nuclear periphery rather than in the nuclear interior (Fig. 1c, panels g,h). To quantify the results, we used PLA between endogenous Repo-Man and transfected GFP:Nup153 or GFP alone. Repo-Man and the GFP:Nup153 PLA signals were more abundant than Repo-Man and the GFP alone and again highly enriched at the nuclear periphery (Fig. 1d, d). Furthermore, PLA signals between endogenous Repo-Man and Nup153 are significantly reduced after Repo-Man RNAi in particular at the nuclear periphery (Fig. 1f). Therefore a close interaction between Repo-Man and Nup153 occurs at the periphery of the interphase nucleus.

To spatially visualize the localization of Repo-Man at the nuclear periphery we used electron microscopy. Since all the available antibodies do not recognize the peripheral pool of Repo-Man in interphase (only during anaphase—see later in the text), we used HeLa cell lines expressing either the N terminus of Repo-Man fused to GFP or GFP alone (Fig. 1g; Supplementary Fig. 1). In this cell line, N terminus Repo-Man shows nuclear localization with particular enrichment at the nuclear envelope (Fig. 2d), in a rim-like configuration similar to the one observed with GFP:Repo-Man full length. This N-terminal region of Repo-Man in fact contains the domain responsible for targeting to the NE[9]. Using antibodies against GFP, we could show that Repo-Man is on the chromatin beneath the NE (Fig. 1g, red arrows), at the edge of the NPC basket (Fig. 1g, black arrow) and a proportion is associated with intra-nuclear bodies in chromatin dense regions, which are possibly related to the proposed role of Repo-Man in heterochromatin formation (see later in the text); further studies however will be required to elucidate their nature (Fig. 1h; Supplementary Fig. 1). Altogether, this data supports the presence of Repo-Man at the nuclear periphery and led us to investigate its possible role on chromatin remodelling in further detail.

Repo-Man accumulates at the nuclear periphery during anaphase where the new pore complex proteins are deposited and there it co-localizes with Nup153 (Fig. 2a); its recruitment at the periphery of the anaphase chromosomes depends on Nup153, in fact Nup153 RNAi prevents the accumulation of endogenous Repo-Man at the chromosome periphery (Fig. 2b,c). We therefore wanted to investigate if its retention at the periphery of the interphase nucleus could also be dependent on the interaction with Nup153. The enrichment of Repo-Man at

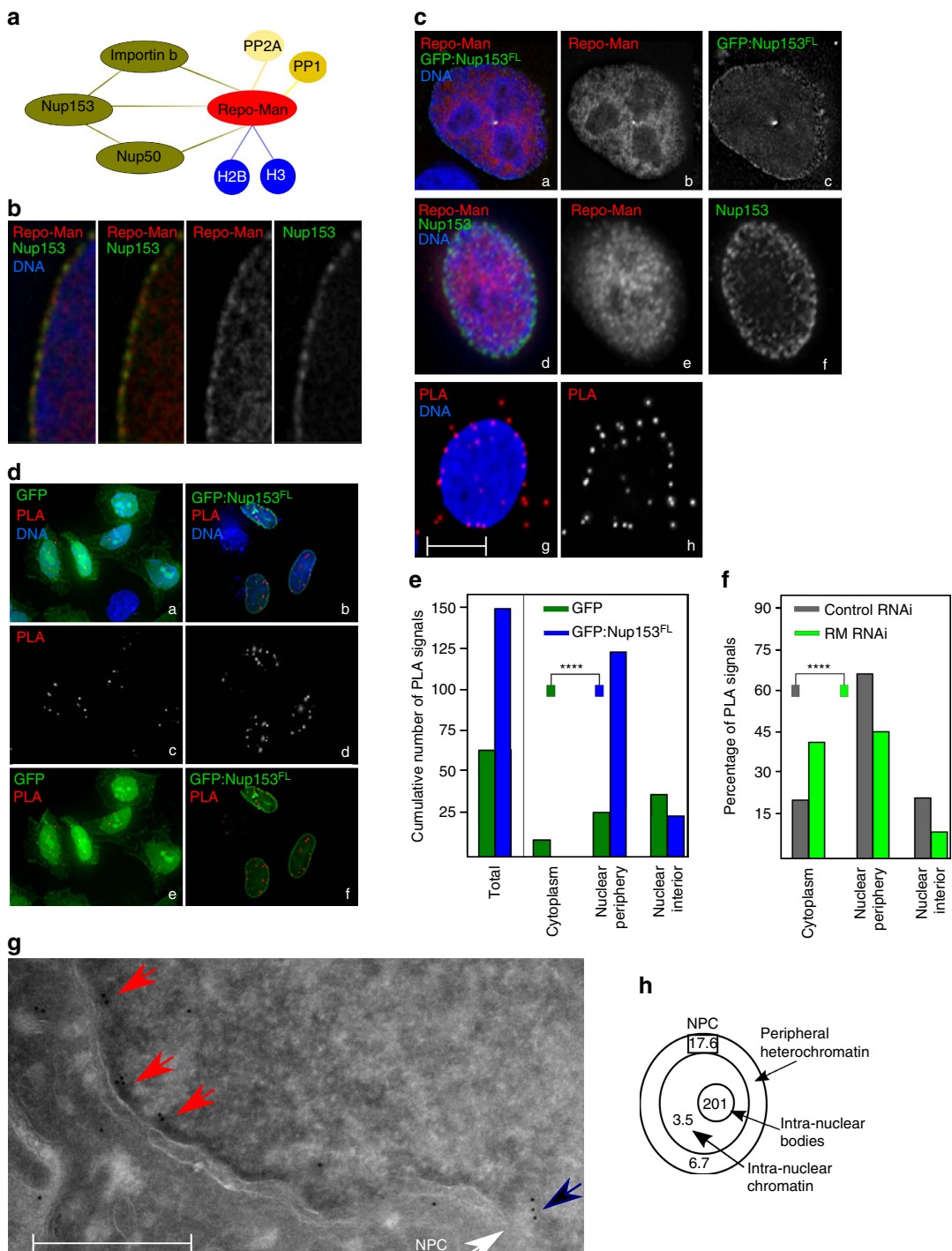

**Figure 1 | Repo-Man is enriched at the periphery of interphase nuclei.** (**a**) Summary of Repo-Man interactors identified in previous studies[9,11,14]. In green interactions with nuclear envelope proteins, blue with histones; yellow phosphatases (PP2A—mitotic exit onset only) and (PP1). (**b**) HeLa cells were transfected with GFP:Repo-Man (red) then fixed and stained for Nup153 (green). (**c**) HeLa cells immunostained for endogenous Repo-Man (red) (a,b,d,e) transfected with GFP:Nup153 (a,c) or co-immunostained for endogenous Nup153 (green) (d,f). Example of PLA signals (red) using Repo-Man and Nup153 antibodies (g,h). Scale bar, 10 μm. (**d**) HeLa cells were transfected with GFP:Nup153 (b,d,f) or GFP alone (a,c,e) and PLA (red) was performed using Repo-Man and GFP antibodies (c,d). (**e**) Quantification and cellular distribution of PLA signals as described in **d** from two independent experiments (Fisher ****P-value < 0.0001). (**f**) Percentage and cellular distribution of PLA signals in Repo-Man (green) or Control RNAi (grey) (Chi-Square, ****P-value < 0.0001). (**g**) Electron Microscopy image of Repo-Man cell line expressing the peripheral N terminus domain fused to GFP. Immuno-electron microscopy was conducted using an anti-GFP antibody. Black arrow shows accumulation at the edge of the NPC (white arrow) and Red arrows show accumulation on heterochromatin adjacent to the nuclear envelope (see Supplementary Fig. 1), scale bar, 500 nm. (**h**) Quantification of the experiment in **g**. Numbers represent the density of labelling in each of the indicated sub-compartments as the number of gold particles μm$^{-2}$ (see materials and methods). Total number of gold particles counted was 1,057.

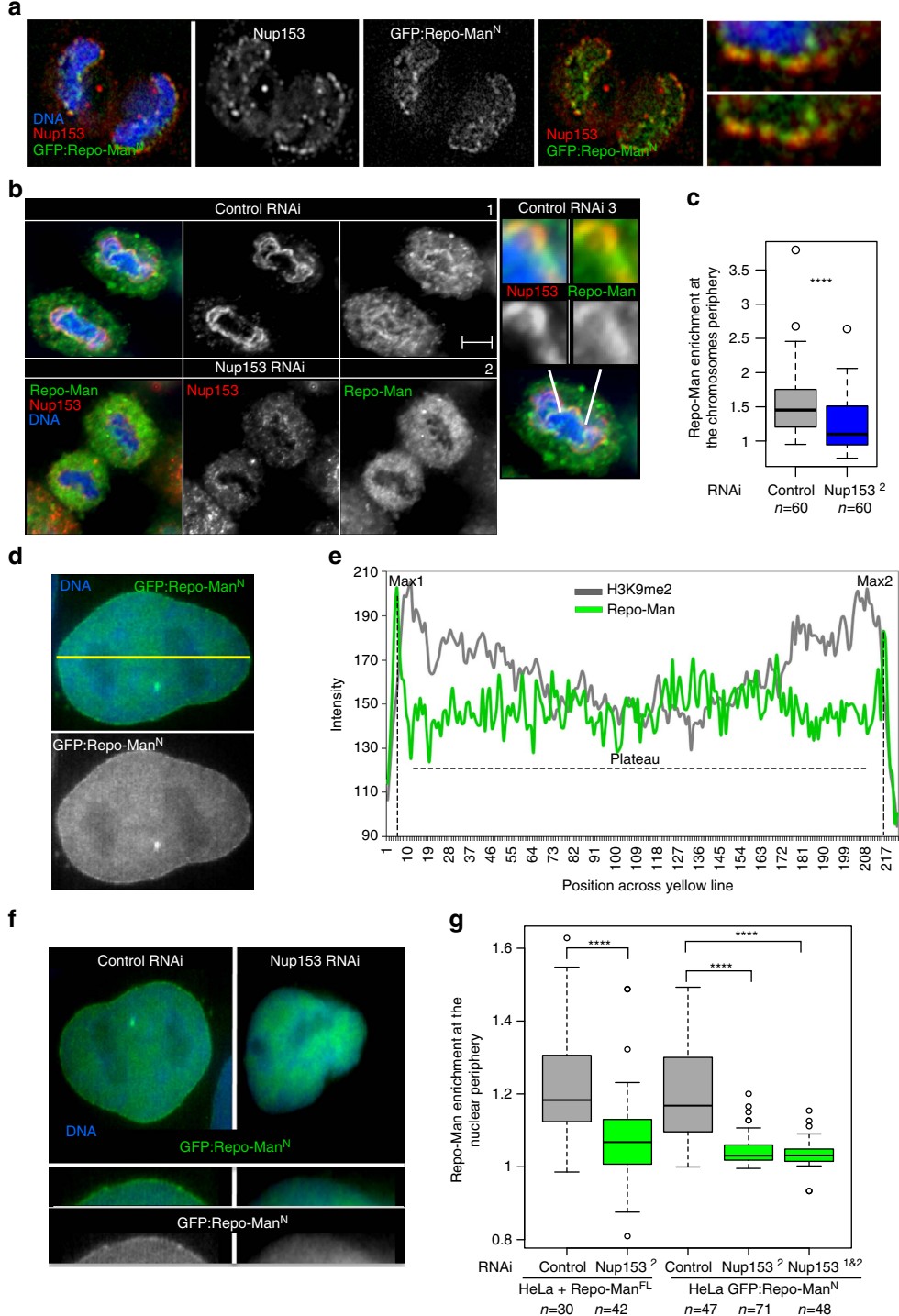

**Figure 2 | Nup153 is necessary for Repo-Man targeting to the nuclear periphery. (a)** Anaphase from HeLa stable cell line expressing GFP:Repo-Man[N] (N terminus Repo-Man), stained for Nup153 (red). Late anaphase showing Repo-Man accumulation at the chromosome periphery and co-localization with Nup153. **(b)** HeLa cells transfected with Control (1) or Nup153[2] RNAi oligo (2) and stained for endogenous Nup153 (red) and endogenous Repo-Man (green). Zoom-in image of Nup153 and Repo-Man at the chromosome periphery in control cells (3). Scale bar, 5 μm. **(c)** Quantification (line profile analysis) of Repo-Man enrichment at the chromosome periphery. **(d)** Interphase nucleus of GFP: Repo-Man[N] HeLa stable cell line showing the nuclear localization of the construct with enrichment at the nuclear periphery. The quantification of Repo-Man distribution was measured as line profile across the nucleus (yellow line) for the experiments in **c** and **g**. **(e)** Profiles of Repo-Man (green) and H3K9me2 (gray) signals across the nucleus. Repo-Man enrichment was measured as the ratio between the average of the two maximum intensity values (Max1 and Max2) by the median of the values in the plateau (**c,g**). **(f)** HeLa cells expressing GFP:Repo-Man[N] were transfected with control or Nup153 siRNA oligos and the GFP profiles were analysed as in **e**. Lower panels are representations of GFP:Repo-Man[N] localization in a section of a nucleus. **(g)** Quantification of Repo-Man enrichment at the nuclear periphery in HeLa cells stably expressing GFP:Repo-Man[N] or transiently transfected with GFP: Repo-Man[FL] *(full-length Repo-Man)* after RNAi with Control oligos (grey bars) or with a single (Nup153[2]) or combination (Nup153[1&2]) Nup153 oligos (green bars). Data in **c** and **g** were analysed with Mann–Whitney test (****$P < 0.0001$), $n$ are depicted in the figures. In box plots in **g**, central line represents the median, box limits are the 25th and 75th percentiles and whiskers extend to 1.5 × interquartile range.

the nuclear periphery, which is slightly external to the peak of the peripheral H3K9me2 marker, was quantified using a line profile analyses (Fig. 2d,e). Depletion of Nup153 indeed leads to the displacement of peripheral Repo-Man from the nuclear lamina, without affecting its nuclear localization (Fig. 2f,g), suggesting that anchoring rather than import is affected by the depletion. The same results were obtained with two different validated oligos against Nup153 (ref. 15) and using both the cell line expressing GFP:Repo-Man N-terminus and the GFP:Repo-Man full-length construct (Fig. 2g).

Therefore, we conclude that Nup153 interacts (directly or indirectly) and recruits Repo-Man thus serving as a platform to enrich or maintain Repo-Man at the nuclear periphery after mitosis.

**Repo-Man regulates post-mitotic heterochromatin assembly.** The nuclear lamina is generally a repressive chromatin compartment enriched for heterochromatin proteins such as HP1 and for repressive histone marks such as H3K9me3 (refs 16–18). HP1 binding to chromatin is dependent on the presence of H3K9me3 and is abolished by phosphorylation of the adjacent serine (S10) by Aurora B in early mitosis[5–7]. Previous work conducted in Neurospora suggested that PP1 could be the molecular effector of this phosho-methyl switch since its depletion causes decreased levels of H3K9me3 (ref. 19). Moreover, we have previously observed that Repo-Man knockdown in HeLa cells leads to increased S10P (ref. 9). Normally, HP1 starts accumulating on chromosomes during mitotic exit and foci become visible in late anaphase[20]. We therefore asked whether Repo-Man was essential for HP1 foci formation in the interphase nucleus.

Repo-Man depletion in HeLa cells leads to a severe decrease in the number and size of HP1 alpha foci (Fig. 3a1 and right panel a–d) that can be rescued by an oligo-resistant version of GFP:Repo-Man (Fig. 3a2). The phenotype is specifically dependent on this particular phosphatase complex since HP1 localization is not affected by depletion of the PP1 subunit SDS22 (Fig. 3a1), previously shown to contribute to the removal of mitotic Aurora B phosphorylations on anaphase chromosomes[21] or, as recently shown, by depletion of another PP1 binding subunit Ki-67 (ref. 22); these data therefore suggest that Repo-Man/PP1-specific substrate dephosphorylations are indeed required for heterochromatin maintenance. Moreover, live cell imaging of GFP:Repo-Man shows that, in cells depleted of Repo-Man, HP1 foci fail to accumulate upon mitotic exit, suggesting that the complex is essential for foci formation (Supplementary Fig. 2a,b). However, Repo-Man RNAi does not decrease the overall level of HP1 (Supplementary Fig. 3a) nor reduces accumulation of lamin B receptor at the nuclear periphery (Supplementary Fig. 3b).

We then wanted to test if enrichment of Repo-Man at a locus was sufficient for HP1 recruitment. To this purpose we used a tethering-recruiting experiment; GFP:LacI:Repo-Man or GFP:LacI were transfected in a DT40 cell line carrying a single integration of LacO repeats[9]. By coupling the LacI/LacO system with immunofluorescence using a series of antibodies against histone modifications and heterochromatin-associated proteins, we have studied their enrichment at the LacO locus.

HP1 is recruited to the LacO array when LacI:Repo-Man but not LacI alone is present (Fig. 3b), and the HP1 accumulation positively correlates with Repo-Man levels (Fig. 3c). HP1 recruitment is dependent on the phosphatase activity of the complex since it is significantly reduced by the Repo-Man RAXA mutant (PP1 non-binding mutant) (Fig. 3b). This therefore suggests that PP1 is necessary for heterochromatin formation. However, tethering PP1 to the locus per se

(via the PP1-binding domain from Ki-67 (refs 23,24)) is not sufficient to restore the level of recruitment achieved by the Repo-Man/PP1 complex (Fig. 3b). Taken together, these experiments clearly indicate that Repo-Man/PP1 complex creates the favourable environment for HP1 recruitment to chromatin. From this picture it emerges that a local balance of active phosphatases is important to maintain the correct level of heterochromatin in cells. It is therefore expected that over-expression of these regulators, either by binding to non-canonical chromatin regions or titrating PP1 away from the bound targeting subunit can produce an abnormal chromatin environment as well; this indeed appears to be the case since Repo-Man overexpression also disrupts the normal accumulation of HP1 foci in interphase nuclei (Supplementary Fig. 2c,d).

Heterochromatin can be accompanied by the presence of H3K9me3 or H3K27me3 propagated by Suv3-9 and the polycomb protein Ezh2 respectively (reviewed in Zhang et al.[25]). Due to the decrease of HP1 foci formation observed upon Repo-Man knockdown we sought to analyse repressive histone post-translational modifications (PTMs) by immunofluorescence. Indeed, Repo-Man RNAi leads to decreased levels of H3K27me2/3 and H3K9me3 (Fig. 3d,e); this is accompanied by an increase in H3K9ac (Fig. 3f) thus suggesting that Repo-Man is necessary to maintain a repressive environment.

Tethering of Repo-Man to a LacO array correspondingly produces accumulation of H3K9me3 and H3K27me2/3 and a decrease in the permissive marker H3K9ac (Fig. 3g). Moreover, H3K9ac levels show an anti-correlation with Repo-Man, suggesting that the presence of Repo-Man is inhibitory for this histone mark deposition whilst Suv3-9 shows the opposite trend (Fig. 3h; Supplementary Fig. 3c,d). All these modifications are indicative of a repressive chromatin status generated by Repo-Man binding that, in this experimental system, is also associated with the appearance of more compacted chromatin as measured by DAPI intensity (Supplementary Fig. 3e).

Altogether, these data provide the first compelling evidence for a role of Repo-Man in heterochromatin formation and maintenance.

**Repo-Man binds to modified H3 tails.** Repo-Man/PP1 is released from the chromatin upon mitotic entry due to a concerted action of CDK-1 (ref. 26) and Aurora B kinases[11]. At anaphase onset, its dephosphorylation by PP2A and PP1 allows the complex to re-localize onto chromatin[8,11]. At this stage, and throughout interphase, the complex has a low turnover[13] suggesting that it is stably associated with chromatin. In fact, 20% of Repo-Man pool is bound to chromatin as shown by cell fractionation experiments (Supplementary Fig. 4b,c), which is in agreement with the amount of Repo-Man immobile fraction observed by FRAP[13].

The chromatin-targeting domain of PP1 has been identified at the C terminus of the protein[9] where aa 890–925 encompass the region necessary for its targeting to chromatin and binding to histones[11].

We next wanted to explore if Repo-Man binds to specific chromatin regions. We first carried out a histone peptide array screening, covering more than 300 histone modifications using the recombinant C-terminus domain of Repo-Man (aa 403–1,023, GST:Repo-Man[CTerm])[9]. GST alone did not provide any signal on the array, while GST:Repo-Man[CTerm] could bind to a subset of histone modifications (Fig. 4a–c). Recombinant Repo-Man preferentially binds to modifications present at lysine 27 of H3, and lysine 20 of H4 (Fig. 4a; Supplementary Fig. 4a). Repo-Man has affinity for the dimethylation, trimethylation and acetylation of lysine 27 of histone H3. These

antagonistic modifications can undergo highly dynamic switches regulated by the abundance of each respective acetyltransferase and methyltransferase (reviewed in Holmqvist and Mannervik[27]). This paradoxical binding can also suggest that Repo-Man associates with different histone modifications via multiple domains. However, Repo-Man does not seem to have high affinity for the other well-established repressive mark H3K9me3.

Interestingly, the array data also shows that the phosphorylation of S28 abolishes Repo-Man binding (Fig. 4b). Since this phosphorylation occurs in mitosis at some but not all the H3 sites

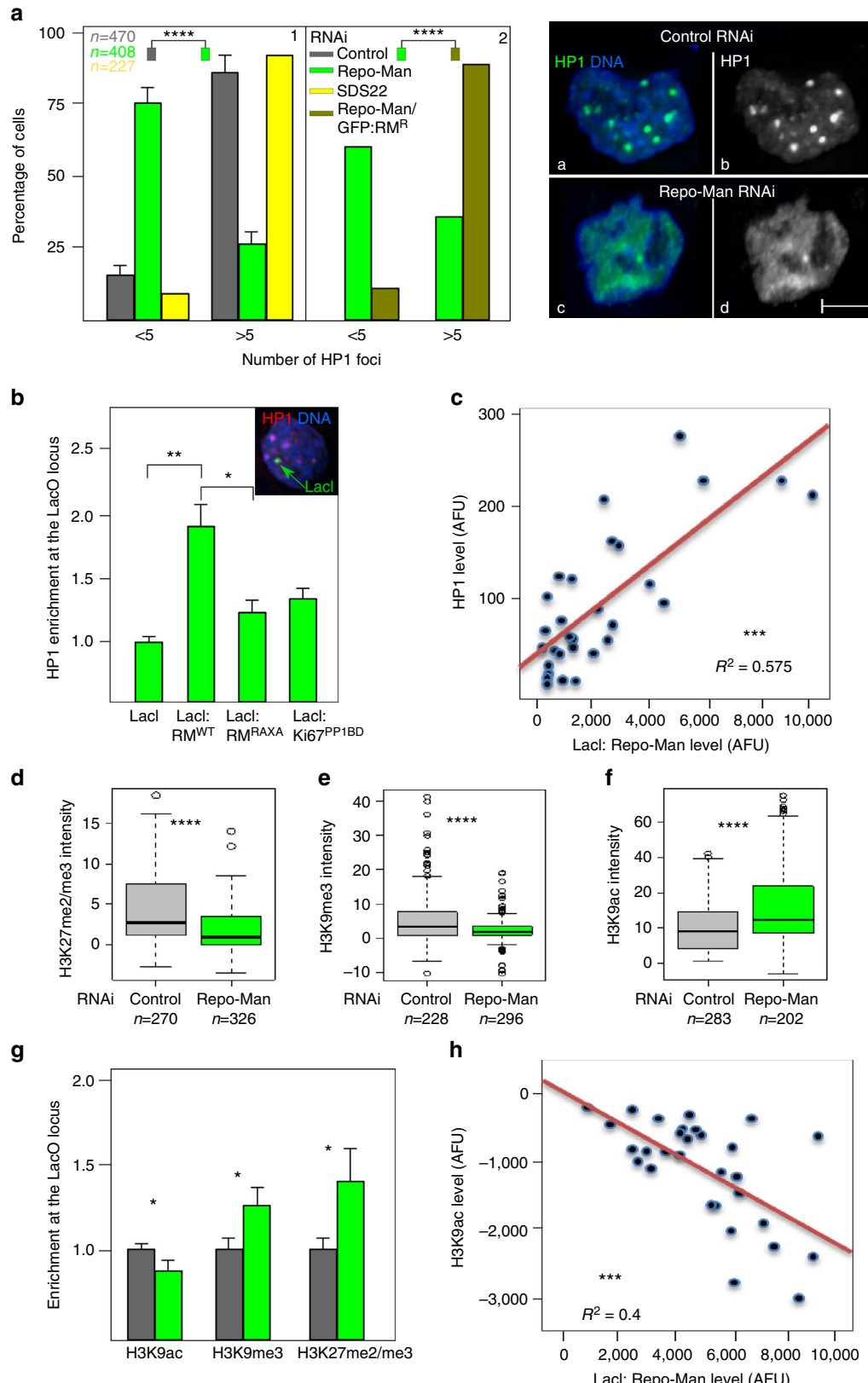

(only 36.5% of H3S28 are phosphorylated in prometaphase, Supplementary Fig. 6d,e), non-phosphorylated H3 sites could be the docking platform for Repo-Man on anaphase chromatin; its recruitment could then direct the dephosphorylation of nearby nucleosomes (see also later in the text and Supplementary Fig. 6b,d,e). This is also in agreement with the analyses on Repo-Man loading onto chromatin during anaphase: it is a progressive accumulation thus supporting a cooperative binding through mitotic exit (Supplementary Fig. 6c).

Repo-Man[1–135], a mutant form that can load on the chromosome periphery but does not appear to be directly targeted onto chromatin[9], does not show any interaction with the histone tails (unmodified or modified) within the array (Fig. 4c). We therefore conclude that only Repo-Man[CTerm] can bind histones *in vitro*. To investigate this histone-binding activity in the context of chromatin, we incubated HeLa nucleosomes with recombinant GST:Repo-Man[CTerm], GST:Repo-Man[1–135] and GST alone (Supplementary Fig. 4e). The presence of histones was only detected in the eluted fraction of GST:Repo-Man[CTerm] (Fig. 4d, Supplementary Fig. 4e).

To verify that Repo-Man is indeed in proximity of H3K27me2/3 *in vivo*, we took advantage of the PLA assay using antibodies against the endogenous Repo-Man and H3K27me2/3 in HeLa cells. The results show positive PLA signals, particularly enriched at the periphery, that are significantly reduced upon Repo-Man knockdown (Fig. 4e,f).

This indicates that the Repo-Man is indeed enriched at chromatin regions containing H3K27me2/3 *in vivo* and in proximity of the nuclear periphery compartment.

**Repo-Man is enriched at subtelomeric regions**. We then investigated where Repo-Man is localized in the genome and the characteristics of the local chromatin environment, in terms of DNA sequences and histone composition.

Previous DamID experiments using a promoter tiling array showed that Repo-Man is not particularly enriched at the promoters of genes[28], therefore we sought to map its binding sites genome-wide. The lack of ChIP grade antibodies for Repo-Man did not allow us to use a ChIP-based approach. We therefore developed a TAG-Proteogenomic approach (Supplementary Fig. 4d); GST-tagged C terminus Repo-Man or GST alone were used as baits to isolate HeLa nucleosomes with high affinity for Repo-Man; after elution, the chromatin bound fraction was used either to separate the histone bands for mass spectrometry analysis (Figs 4d and 5a) or to extract DNA for sequencing (HiSeq) (Fig. 5b–f) (see schematics in Supplementary Fig. 4d). This approach provided us with unbiased information on both the chromatin flavour in terms of histone variants and modifications as well as the genomic binding regions of Repo-Man. The mass spectrometry

data analyses of two independent repeats show that Repo-Man binds preferentially to chromatin containing the H2A variant H2AZ and the H3 variants H3.2 or H3.3.

Histone variants have different functions in chromatin. For example, H3.3 is incorporated in a replication-independent manner and is found at active regions[29] but also at silenced regions such as pericentromeric regions and telomeres[30].

H2AZ, which comprises only 10% of total H2A (ref. 31) seems to be necessary for telomeric repression in yeast[32] and to be upstream of H3K9me3 and HP1 recruitment in drosophila[33].

Since Histone H3 sequences are almost identical it is not surprising that the vast majority of PTMs found are shared amongst the three H3 variants (Fig. 5a). Nevertheless, we have identified some H3 PTMs that are specific for H3.3 in Repo-Man-associated chromatin. In particular, the H3K27me2/K36me2 marks were shown to co-exist on the same H3 tail and being dependent on PRC2 (ref. 34). Repo-Man bound chromatin seems to be enriched for several well established repressive marks (K9me1, K27me1, K27me3) and others (K79me1, K79me2 or K115me1) whose function are less understood (Fig. 5a). Intriguingly, K27ac is found associated with H3.1 and H3.2 whereas K27me3 with H3.3 suggesting that Repo-Man/PP1 could interact with two intrinsically different nucleosome structures.

To identify which regions within the genome Repo-Man is capable of binding to, we used the eluted chromatin from the experiment described before and analysed the DNA using deep sequencing (Supplementary Fig. 4d).

Repo-Man is found distributed on all the chromosomes, as expected from the known cell biology of the complex. A significant 9-fold enrichment of Repo-Man binding sites was observed at subtelomeric regions of several chromosomes (Fig. 5b,c). This is consistent with the mass spectrometry results.

Characterization of Repo-Man binding sites reveals significant enrichment for RefSeq genes and exons (Fig. 5d). Although in coverage very little is found on TSS even when the window comprises a 2Kb region around the TSS, 2 and 15% respectively (not shown), these are still significant. This is in line with the DamID promoter tiling array experiments[28], where few promoter hits were found for Repo-Man when compared with other PP1 targeting subunits; an example is *MEST* gene detected in our and the published data set (Supplementary Fig. 4f). Repo-Man is over-represented at CpG islands. CpG islands are often associated with active gene promoters but they were also found at promoters of developmentally regulated genes and repressed by polycomb group of proteins (reviewed in Deaton and Bird[35]).

We then analysed Repo-Man accumulation relative to the presence of combinations of histone modifications with

**Figure 3 | Repo-Man is necessary and sufficient to establish a heterochromatic environment.** (**a**) Quantification of HP1 alpha foci after immunostaining of HeLa cells depleted of Repo-Man (green) or SDS22 (yellow) (1). Rescue of the HP1 foci numbers is achieved by a Repo-Man:GFP oligo-resistant construct in a Repo-Man siRNA background (brown) (2). Chi-Square (****$P < 0.0001$). Typical image of HP1 foci in a control (a,b) or Repo-Man (c,d) RNAi. Scale bar, 5 μm. (**b**) DT40 cells containing a LacO array inserted in a single locus were transfected with GFP:LacI, GFP:LacI:Repo-Man(RM), GFP:LacI:Repo-Man(RM)[RAXA] and the PP1 binding domain GFP:LacI:Ki67[PP1BD]. Cells were fixed and stained with HP1 antibody (representative image shown in the inset). The enrichment was calculated as a ratio between the intensity at LacI spot (green arrow in the inset), and a random nuclear spot. (**c**) Correlation between the accumulation of GFP:LacI:Repo-Man(RM) at the LacO array and HP1 from the experiments in **b**, linear regression. (**d–f**) Intensities of H3K27me2/3 (**d**), H3K9me3 (**e**) and H3K9ac (**f**) staining in fixed HeLa cells after control (gray) or Repo-Man (green) RNAi. Cell numbers are depicted in the figure. Data sets were analysed with Mann–Whitney test between three replicates (****$P < 0.0001$). In box plots, central line represents the median, box limits are the 25th and 75th percentiles and whiskers extend to 1.5 × interquartile range. (**g,h**) DT40 cells containing a LacO array inserted at a single locus were transfected with GFP:LacI (grey) or GFP:LacI:Repo-Man (green). Cells were fixed and stained with antibodies against H3K9ac, H3K9me3 and H3K27me2/3. The signal intensity levels were measured as described in **b**. (**h**) Correlation between GFP:LacI:Repo-Man(RM) enrichment at the LacO array and the levels of the active H3K9ac, linear regression. Stars indicate *t*-test unless stated otherwise (*$P < 0.05$, **$P < 0.01$, ***$P < 0.001$ using two-three replicates). Error bars in **a**, **b** and **g** represent s.d.

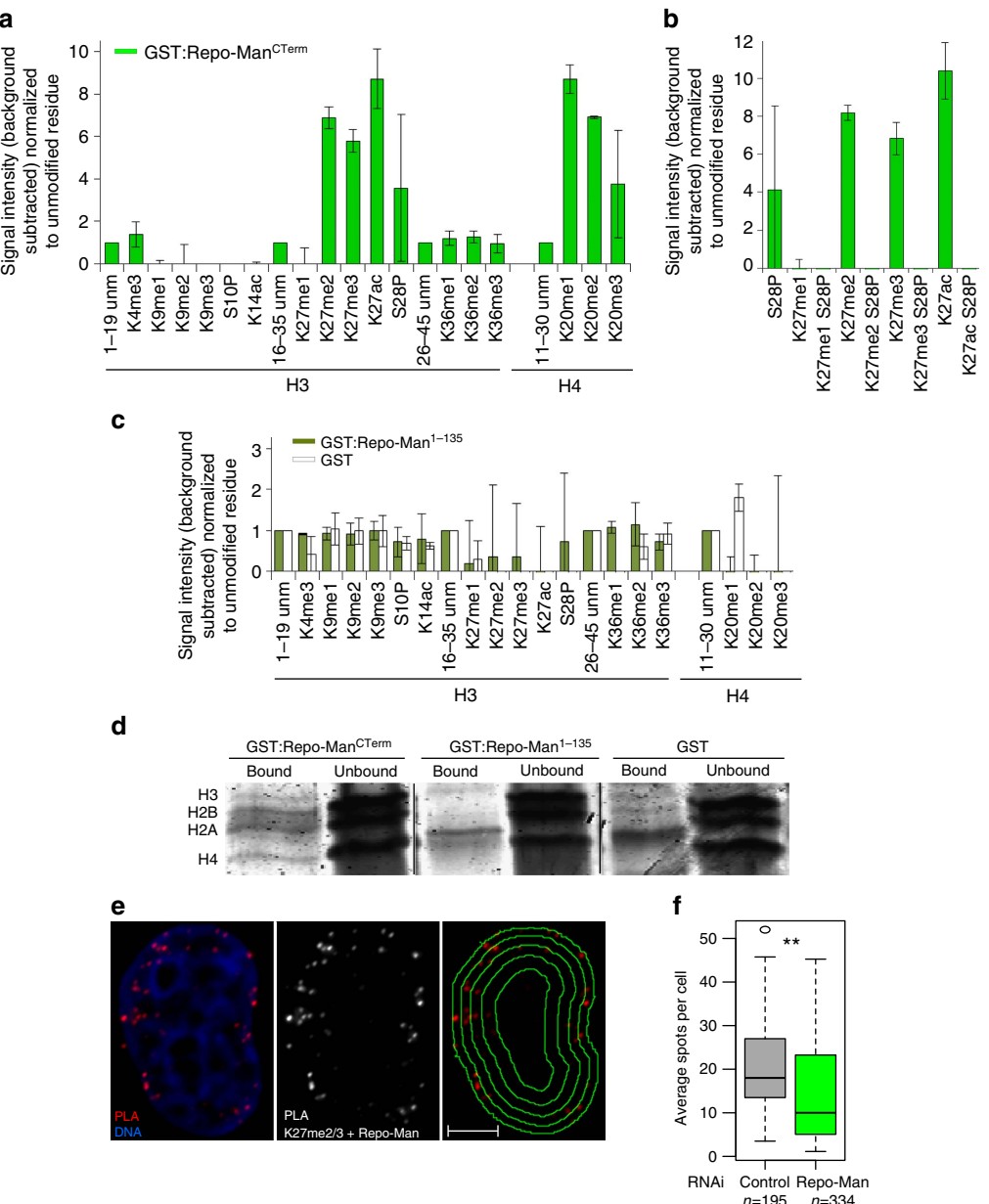

**Figure 4 | Repo-Man interacts with modified histone H3. (a)** Recombinant GST tagged Repo-Man (C terminus domain) was incubated with a histone peptide array (Active Motif). The signal intensity was detected with an anti-GST antibody and quantified by LICOR. Preferential interactions of GST:Repo-Man[CTerm] (C-terminus domain) are shown between two biological replicas. **(b)** Repo-Man has less affinity for peptides containing modified K27 residue (either methylated or acetylated) if the adjacent S28 is phosphorylated (S28P). **(c)** The histone peptide array was incubated with recombinant GST:Repo-Man[1–135] (dark green bars) or GST alone (white bars) and signals detected with an anti-GST antibody as in **a**. Error bars represent s.e.m. between two arrays. **(d)** InstantBlue staining of GST alone, GST:Repo-Man[CTerm] or GST:Repo-Man[1–135] proteins incubated with HeLa nucleosomes (bound and unbound fractions are shown). **(e)** Endogenous Repo-Man and H3K27me2/3 interactions in interphase detected by PLA. Panel on the right shows the overlay of the PLA signals with the nuclear erosion script[69]. Scale bar, 5 μm. **(f)** Counts of PLA signals in control and Repo-Man RNAi in two replicates as described in **c**. P-value was calculated using Mann–Whitney test (**P < 0.01). n is depicted in the figure. In box plots, central line represents the median, box limits are the 25th and 75th percentiles and whiskers extend to 1.5 × interquartile range.

particular attention to the ones identified in the peptide array or in our mass spectrometry data sets (Fig. 5e). The co-existence of H3K27me3 and H3K4me3 is the classical bivalent mark for developmentally regulated genes but it is also present in differentiated cell lines[36]; Repo-Man is significantly enriched at sites encompassing these markers.

H3K27me3 together with H3K9me3 has been found in 48% of drosophila polythene chromosomes[37] but also in human cells and evidences indicate that the modified H3K27me3 and H3K9me3 reinforce heterochromatin establishment through

HP1 alpha associations[6]. Despite the high overlap of Repo-Man with H3K9me3 and H3K27me3, this is not statistically significant, possibly pointing at the high representation of these marks in differentiated cells[38].

Repo-Man is also highly enriched in H3K79me2 and H3K4me3 or H3K27me3 marked chromatin. The role of H3K79me2 in the epigenetic landscape is still not fully understood; although most studies seem to indicate a role in transcription, H3K79me2 has also been associated with Swi6 (HP1)[39] and it has been postulated to occupy bivalent

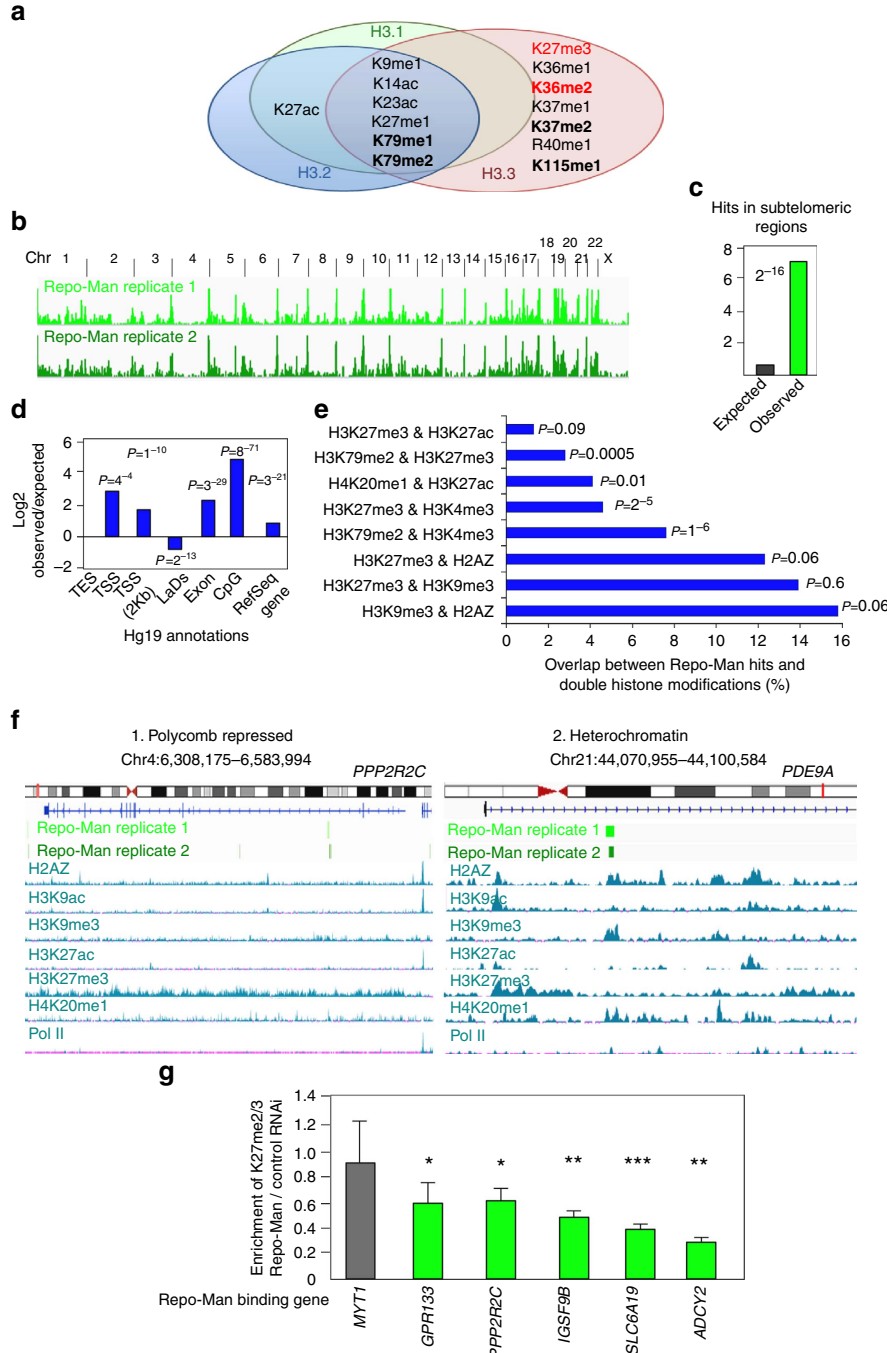

**Figure 5 | Repo-Man associates with repressive histone modifications and subtelomeric regions.** Eluted fractions of GST:Repo-Man and GST alone incubated with nucleosomes (as in Fig. 4d) were analysed by Mass Spectrometry (**a**) or the DNA was extracted and sequenced by Illumina HiSeq (**b**-**f**). (**a**) Histone H3 PTMs identified in the GST:Repo-Man fraction only. In bold are modifications identified in all the replicate experiments and in red are coexistent PRC-dependent histone modifications found in Jung *et al.*[34] (**b**) Overview of GST:Repo-Man binding sites genome-wide in two replicates. (**c**) Repo-Man hits at subtelomeric regions are higher than expected by chance. (**d**) Annotation of Repo-Man hits according to gene features or lamina association[71] (Fisher *P*-values). TES: transcription end site; TSS: transcription start site; LADs: lamina associated domains. (**e**) Overlaps between Repo-Man hits and double histone modifications extracted from HeLa ENCODE data sets for H3K27ac, H3K4me3, H3K79me2, H3K27me3, H2A, H3K9me3 and H4K20me1 (Fisher *P*-values). (**f**) Single gene profiles of Repo-Man target genes *PPP2R2C* (1) and *PDE9A* (2) classified as polycomb repressed and heterochromatin associated (H3K9me3) respectively by the software ChromHMM[43]. The chromosomes and the position of the gene (red line) are shown along with the representation of the genomic sequence (lines/squares are exons). Repo-Man binding sites distribution is shown for two independent data sets (light and dark green). Positioning of histone marks along the genomic window were extracted from the UCSC in HeLa cells (H2AZ, H3K9ac, H3K9me3, H3K27ac, H3K27me3, H4K20me1 and S2-PolII), reads in *y* axis = 50. (**g**) H3K27me2/3 ChIP on chromatin from control and Repo-Man RNAi cells. Repo-Man RNAi enrichment is expressed over Control RNAi enrichment, calculated relatively to input DNA using same amount of DNA in PCR. Error bars represent s.e.m. *t*-test was applied. (**P* < 0.05, ***P* < 0.01, ****P* < 0.001).

genes together with H3K4me3 (often on the same nucleosomes[40]) and H3K27me3 (ref. 41).

The overrepresentation of Repo-Man binding sites at regions containing H4K20me1 and H3K27ac is not surprising since these marks are associated with CpG island promoters[42].

Two typical gene profiles, *PPP2R2C* and *PDE9A*, are shown in Fig. 5f together with Repo-Man occupancy and histone profiles. According to the bioinformatics tool ChromHMM[43], *PPP2R2C* is defined as polycomb repressed (H3K27me3 positive) whereas *PDE9A* is defined as heterochromatin (H3K9me3 positive). These genes are also characterized by the presence of H2AZ and absence of PolII (Fig. 5f and other examples in Supplementary Fig. 4g).

We next sought to explore the functional relationship between H3K27me2/3 occupancy and Repo-Man. To this purpose we have selected polycomb genes containing Repo-Man-binding sites and performed ChIP on Control or Repo-Man RNAi-treated cells (Fig. 5g). Across this panel of genes, Repo-Man RNAi reduces the accumulation of H3K27me2/3 thus reinforcing Repo-Man role in the maintenance of this repressive chromatin.

**Repo-Man regulates chromatin positioning and gene expression.** We have so far shown that Repo-Man sustains a repressive environment, it is enriched at subtelomeric regions and is important for chromatin organization at the nuclear periphery. To examine the biological implications of these findings we used a HT1080 cell line containing LacO arrays integrated in the 13q22 and expressing LacI:GFP[44]. Within this region, there are Repo-Man binding sites and, importantly, this locus is found to localize at the nuclear periphery (Fig. 6a). Upon Repo-Man RNAi (Fig. 6b) we could observe repositioning of the peripheral chromosomal 13q22, with the locus moving towards the interior of the nucleus. Positioning of the locus seems to be dependent on PP1 since overexpression of Repo-Man RAXA (a dominant-negative form that does not bind PP1) shows a similar trend (Fig. 6b). However, Repo-Man depletion did not change the positioning of the nucleolar-associated locus on chromosome 13p (Supplementary Fig. 5a,b). We next explored the impact of Repo-Man on an endogenous locus, the subtelomeric region of chr14 (Supplementary Fig. 5c). Using FISH with a PAC mapping to this region (CTC-820M16 (ref. 45)) we show that the locus moves to a more central location upon Repo-Man RNAi in HeLa cells (Fig. 6c,d). Together, these results support a role for Repo-Man in maintaining a subset of subtelomeric regions at the nuclear periphery.

This further evidence also suggests that the Repo-Man/PP1 complex at the nuclear periphery is important to maintain heterochromatin features necessary for the spatial organization of chromatin within the nucleus.

Due to the fact that chromosome positioning at the periphery is closely linked to transcriptional repression and our new findings of Repo-Man being associated with several polycomb repressed genes, we tested if Repo-Man dosage would affect the expression of some of these telomeric-located genes. We selected five genes based on their telomeric positioning, enrichment for H3K27me2/3 and association with Repo-Man (*ADCY2*, *GRP133*, *SLC6A18*, *PPP2R2C* and *SLC6A19*). The expression profile of these genes, assessed by qPCR upon Repo-Man RNAi, shows an increase in their expression consistent with Repo-Man/PP1 playing an active role in maintaining a repressive environment at these telomeric loci (Fig. 6g).

Because we have identified Nup153 as having a critical role in recruiting and maintaining Repo-Man at the periphery, we tested if depletion of Nup153 itself would affect the peripheral chromatin organization. We have therefore analysed the

enrichment of H3K27me2/3 and H3K9me3 at the nuclear periphery after Nup153 RNAi; in this condition both markers are decreased at this nuclear compartment (Fig. 6e,f) and more importantly, a selection of Repo-Man bound subtelomeric genes became de-repressed (Fig. 6g).

**Repo-Man dephosphorylates H3S28.** We have previously shown that Repo-Man/PP1 is essential for the dephosphorylation of H3S10 during mitotic exit however the phosphatase for the H3S28 site is not known. H3S28 is phosphorylated both in mitosis (by Aurora B) and in interphase (by MSK1) in response to stress[46]. This phosphorylation helps to modulate the binding of PRC2 and the expression of polycomb-regulated genes (reviewed in Sawicka and Seiser[47]). We therefore tested if Repo-Man/PP1 dephosphorylates H3S28 for which a phosphatase has not been identified.

To this purpose we used two different approaches. First, we overexpressed a hyperactive Repo-Man mutant TA3 (previously characterized alongside with its effects in H3T3 dephosphorylation[9]) and tested the phosphorylation levels of H3S28 in early mitosis; indeed Repo-Man TA3 can induce premature histone dephosphorylation of H3S28 (Fig. 7a). Second, we depleted Repo-Man in HeLa cells and analysed the H3S28 phosphorylation levels in anaphase and cytokinesis: in Repo-Man depleted cells a significant level of H3S28 is retained compared with the controls (Fig. 7b,c). Moreover, mitotic chromosomes of Repo-Man depleted cells show a higher level of H3S28 phosphorylation (Supplementary Fig. 6a). All together this data suggests that H3S28 is a substrate of Repo-Man/PP1 at least during mitotic exit. However, due to Repo-Man being bound to some chromatin regions during interphase and the fact that some polycomb-repressed genes increase expression upon Repo-Man RNAi, we can speculate that Repo-Man/PP1 could potentially dephosphorylate H3S28 in interphase as well.

These results clearly identify Repo-Man/PP1 as a key chromatin-linked phosphatase complex essential for modulating the levels of S10 and S28 phosphorylations, which might drive phospho/methyl switches through HP1 and PRC2 binding (Fig. 7d).

## Discussion
Repo-Man/PP1 complex has been shown to have important functions both in early mitosis and during mitotic exit. One of the established roles for this complex is reverting the mitotic phosphorylation events on histone H3, T3/S10 (ref. 48), and here we show that dephosphorylates the S28 residue as well. Beside this catalytic activity, Repo-Man is also an important factor necessary for the timely re-organization of the nuclear envelope during mitotic exit. However, this complex is not degraded once mitosis is over but remains stably linked to chromatin in interphase until the following division. In the present study we found that a fraction of Repo-Man is enriched at the nuclear periphery where it is maintained via an interaction with Nup153, associates with heterochromatin marks and is essential for the peripheral localization, epigenetic environment maintenance and expression of a subset of subtelomeric and polycomb-regulated genes.

Previous studies demonstrated that Repo-Man interacts with Importin β in anaphase and that its depletion leads to deformed G1 nuclei. After cell division, it remains associated with chromatin[13], and a fraction is bound to the nuclear periphery. Nup153 has been previously shown to co-purify with Repo-Man in chicken and human cells[9,14] but the function of this interaction remained unknown. Here we have demonstrated that this interaction is important to stabilize

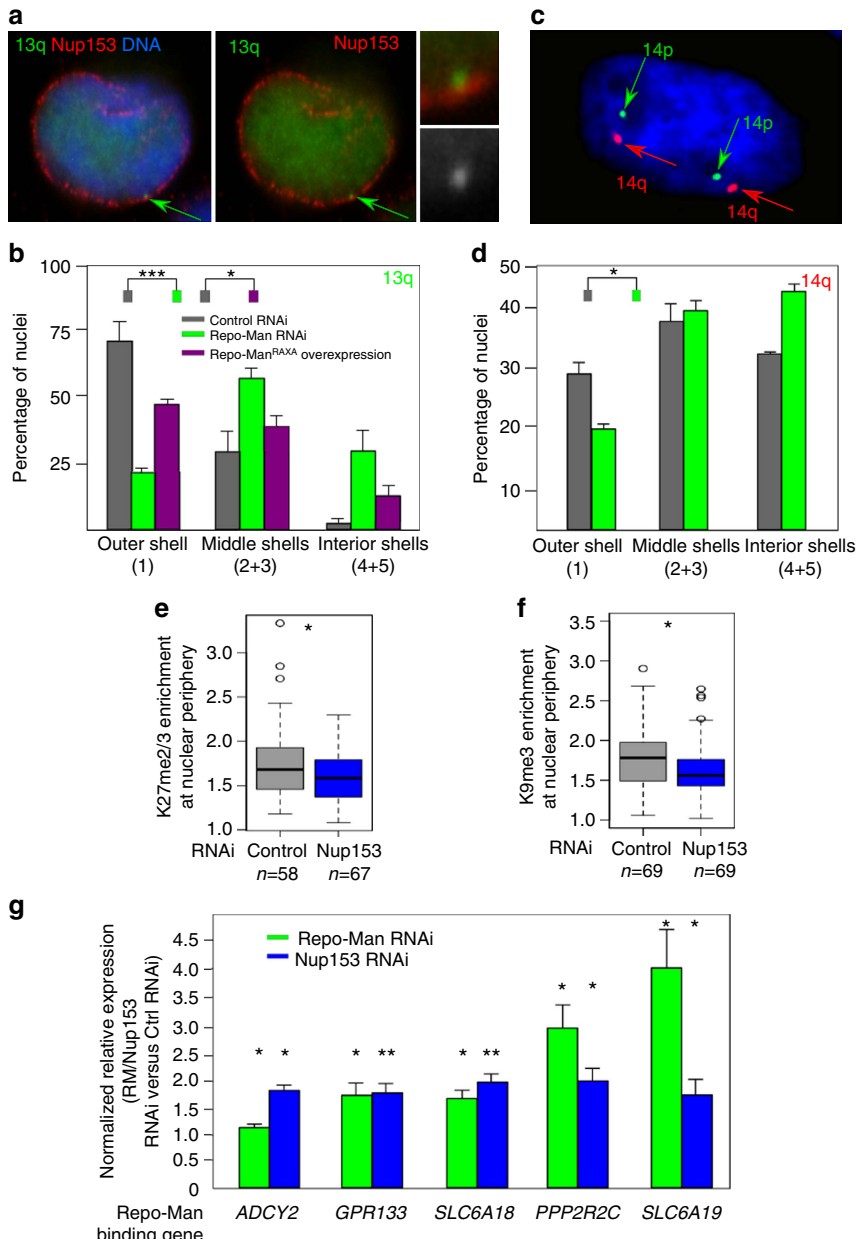

**Figure 6 | Repo-Man depletion affects chromosome positioning.** (**a**) HT1080 cells containing a LacO array inserted at 13q22 and expressing GFP:LacI were fixed and stained for Nup153. The arrow indicates the integration site. (**b**) Position of the chr13q22 was measured using an erosion script software[69] across five concentric shells (1—most outer shell to 5—most inner shell) after RNAi with control or Repo-Man oligos or transiently transfected with the dominant-negative Repo-Man[RAXA] mutant. (**c**) 3D FISH with probe CTC-820M16 (red signal) mapping to the subtelomeric region of chromosome 14 performed on HeLa cells. (**d**) Quantification of spots location described in **c**, using the erosion script software. (Fisher test, *$P<0.05$, **$P<0.01$, ***$P<0.001$ using two-three replicates). (**e**) Enrichment of H3K27me2/3 at the nuclear periphery after Nup153 RNAi. (**f**) Enrichment of H3K9me3 at the nuclear periphery after Nup153 RNAi. Enrichment was calculated as in Fig. 2e. Mann–Whitney test (*$P<0.05$), n is depicted in the figure. In box plots in **e** and **f**, central line represents the median, box limits are the 25th and 75th percentiles and whiskers extend to 1.5 × interquartile range. (**g**) Differential expression of telomeric genes bound by Repo-Man between control and Repo-Man (green) or Nup153 (blue) RNAi. Delta–delta-CT method was used and normalized for *GAPDH*. Error bars = s.e.m. between three replicates. *t*-test was used for statistical analysis (*$P<0.05$, **$P<0.01$).

the pool of Repo-Man at the nuclear periphery of the newly-formed nuclei and point to an extended function for Nup153 involved also in tethering chromatin-associated proteins, distinct from interactions with nucleoporins and import proteins necessary for the normal function of the NPC.

Using electron microscopy we were able to show that Repo-Man lies right outside the NPC, it sits on the dense chromatin at its boundary as well as underneath the nuclear lamina within patches of heterochromatin, pointing that

Repo-Man bound chromatin is indeed different from the active type of chromatin commonly associated within the nucleopore basket[49]. Interestingly, recent investigations in ES cells have also shown enrichment of polycomb repressive chromatin associated with Nup153 (ref. 50).

Repo-Man contributes to the re-formation of HP1 foci after division and its depletion coincides with decreases in heterochromatin marks such as H3K27me3 and, to a lesser extent, H3K9me3. When Repo-Man is artificially tethered

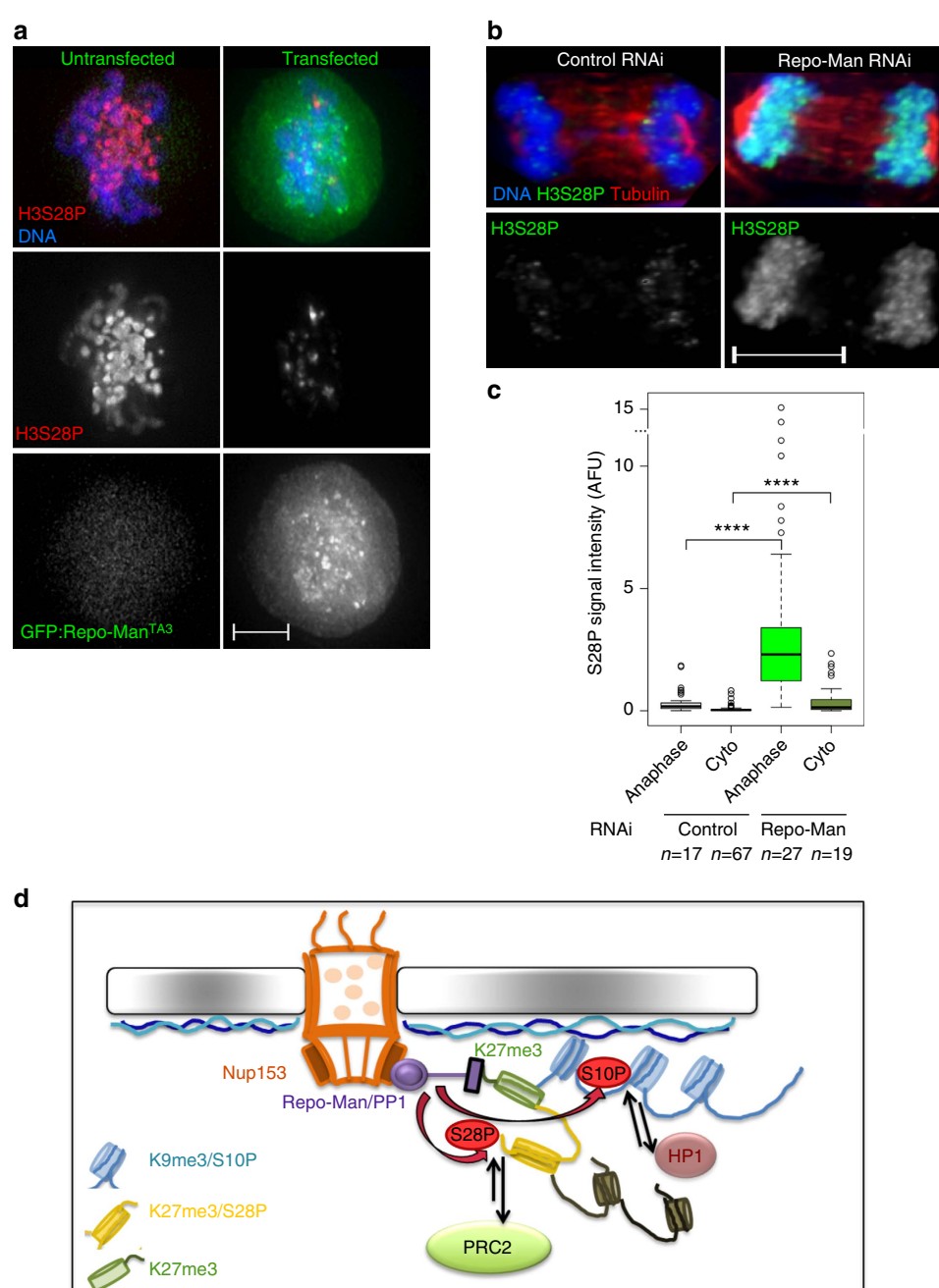

**Figure 7 | Repo-Man dephosphorylates S28 and regulates a phospho-switch necessary for heterochromatin maintenance. (a–c)** Repo-Man dephosphorylates H3S28P. **(a)** HeLa cells overexpressing GFP:Repo-Man[TA3], which prematurely associates with chromatin, were fixed and stained for H3S28P. Scale bar, 5 μm. **(b,c)** HeLa cells after control or Repo-Man RNAi were fixed and stained for H3S28P and tubulin. The intensity of H3S28P was measured in anaphase and cytokinesis (cyto) in two independent replicates **(c)**. Scale bar, 10 μm. Mann–Whitney test (****$P < 0.0001$), $n$ is depicted in the figure. In box plots in c, central line represents the median, box limits are the 25th and 75th percentiles and whiskers extend to 1.5 × interquartile range. **(d)** Model: Repo-Man associates with modified Lysine 27 when the adjacent S28 is not modified. Through dephosphorylations of the nearby serine 10 and 28 it regulates HP1 enrichment and potentially the maintenance of H3K27 methylation respectively. These processes may contribute to the establishment and/or maintenance of a repressive environment. At the periphery, the position of this chromatin environment is also locally maintained via the interaction between Repo-Man and Nup153.

to a locus (using the LacI:LacO system), repressive histone marks accumulate at this region. All these evidences strongly suggest that Repo-Man is necessary and sufficient to generate and maintain a repressive chromatin environment within the nucleus. In fact, the nuclear periphery is enriched in repressive chromatin and the pool of Repo-Man localized at the periphery could be important to maintain this nuclear environment.

In this respect, the important question is: how does Repo-Man bind to chromatin and where?

Repo-Man binds progressively to chromatin after anaphase onset[8] through its histone-binding domain localized to the C terminus of the protein[9]. Binding of PP1 and PP2A to Repo-Man allows its dephosphorylation and targeting to chromatin[11]. Here we have identified that Repo-Man has affinity for a subset of histone modifications including

H3K27 (me2/3 or ac) that could represent its docking sites at anaphase onset in nucleosomes voided of S28 phosphorylation; in fact only 36.5% of S28 is phosphorylated in mitosis. Because of this finding and with the knowledge that Repo-Man is involved in the dephosphorylation of histone H3, we hypothesized that Repo-Man could also target S28 for dephosphorylation, for which thus far no phosphatases had been identified. Our study in fact indicates that this is the case.

These findings place Repo-Man/PP1 at the centre of the phosphorylation switches occurring on histone H3 critical for the M/G1 transition but also suggest that this complex could be involved in signal transduction in interphase, for example opposing the activity of MSK1 and PLK1 (refs 51,52). The latter is a very interesting avenue that will be interesting to explore further in a well-characterized model system.

Our peptide array analyses identified the binding to H3K27me2/3 but also to H3K27ac and H4K20me1/2. A dynamic interplay between H3K27ac and H3K27me3 has been shown in ES cells. Depending on the levels of the respective enzymes, developmentally regulated genes seem to transit between an active, marked by H3K27ac, and poised, marked by H3K27me3 state[53]. A rapid transition between these states would possibly require that chromatin readers could recognize opposing modifications perhaps with different affinities. Another possibility is that Repo-Man C-terminus domain interacts with positive and negative histone markers via different subdomains.

H4K20me1 and H4K20me2 are widely present in the genome and H4K20me3 is deposited by Suv4-20h2, which interacts with H3K9me3 and HP1 (refs 54–56). It might be the case that Repo-Man binds the bookmarked H4K20me1/2 for facilitating a later conversion into the repressive H4K20me3 state. Further studies will be necessary to analyse in detail the biological significance of Repo-Man and H4K20me binding. The presence of combinations of these modifications within the same or adjacent nucleosomes could increase and tighten the binding of Repo-Man to a specific chromatin region. Nucleosome reconstitution experiments will address these important questions in the future.

Chromatin containing the H2A variant H2AZ seems to be preferentially bound by Repo-Man. Interestingly, H2AZ is associated with polycomb-repressed genes and its loss reduces PRC2 occupancy levels in ES cells[57].

Repo-Man binds chromatin characterized by a repressive histone code and is enriched at subtelomeric regions. Telomeres are enriched for the histone variant H3.3 and often found associated with the repressive markers H3K9me3, H4K20me3 and HP1 at subtelomeric regions[58] and H2AZ in yeast[32]. A high percentage of Repo-Man binding sites overlap with H3K9me3 and H3K27me3 and in conjunction with H2AZ. The first two histone modifications are markers of repression (H3K9me3 and H3K27me3) and the latter demarks insulator regions (H2AZ). Moreover, H3K9me3 and H3K27me3 can co-associate and they have been found together with H2AZ in facultative heterochromatin bound by lamin A/C, thought to be more dynamic[59].

Very little Repo-Man is found at promoters, as previously shown using a DamID approach[28], although its presence in these regions is higher than expected. Interestingly, Repo-Man has a large portion of binding sites at CpG islands. CpG islands are usually found in promoters of active genes and studies in ES cells found both H3K4me3 and H3K27me3 at CpG islands bound by PRC2 (ref. 35).

The functional relevance of Repo-Man targeting to telomeres and its presence at the nuclear periphery has implications in genome organization. In fact we have shown that subtelomeric regions of chr13 and chr14 move away from the periphery

in Repo-Man depleted cells and the expression of telomeric genes is elevated upon RNAi. These changes are not simply explained by cell-cycle arrest/defects caused by Repo-Man knockdown since previous studies show that Repo-Man depletion does not cause major cell cycle changes in HeLa cells[8]. It has been suggested that telomeres are tethered to the periphery in late anaphase during the process of nuclear envelope reassembly[60]. Repo-Man is crucial for the nuclear assembly process as well[9], suggesting that the role of Repo-Man in telomere organization may begin at these early stages of nuclear formation. Moreover we have provided the first evidence of a role of Nup153 in organizing the peripheral chromatin via Repo-Man.

In the present study we have also shown that Repo-Man not only counteracts the S10 (ref. 9) phosphorylation but also the S28 during mitotic exit; this role could also persist in interphase at specific sites. Multiple kinases, including RSK2, MSK1/2, PIM1 and IKKα, have been shown to directly phosphorylate H3 thereby indicating that H3 phosphorylation is a critical step in signal transduction to the chromatin/transcriptional regulatory machinery (reviewed in Baek[61]). Stress induction of MSK1 can re-activate the polycomb-silenced α-globin gene via H3S28 phophorylation[62] and gene activation during ES cells differentiation through dissociation of PRC[63]. On the other hand, H3S28P could also trigger more permanent changes in the epigenetic landscape. A methyl/acetylation switch on the lysine 27 has been proposed in a luciferase reporter where MSK1 phosphorylation of S28P leads to K27 acetylation coupled with reduction of K27me3 and of polycomb group of proteins binding at the reporter[62]. Previous models suggest that phosphorylation of S28 through stress activated kinases underlies the methyl/acetyl switches regulating the nearby K27, however, the nature of the counteracting phosphatase is not known[64]. Since we detect a decrease of H3K27me2/3 after Repo-Man RNAi at specific loci where Repo-Man is bound, we suggest that Repo-Man could be important in mediating the acetyl-methyl switch, through dephosphorylation of H3.

Collectively our data shows that Repo-Man/PP1 facilitates a particular chromatin environment in the daughter cells and, by doing so, this complex contributes to shape nuclear chromatin structure and organization in interphase. This represents the first study suggesting Repo-Man/PP1 complex as epigenetic regulator. As can be predicted by this model, a phosphatase with such a role should be maintained at highly controlled levels and alterations of its dosage may have drastic consequences for gene organization and expression that might arise in disease scenarios like cancer.

## Methods

**Cell culture, cloning and transfections.** HT1080 and HeLa cells were grown in DMEM supplemented with 10% fetal bovine serum (FBS) and 1% Penicillin–Streptomycin (Invitrogen Gibco) at 37 °C with 5% CO2.

DT40 cells carrying a single integration of the LacO array[26] were cultured in RPMI1640 supplemented with 10% FBS and 1% chicken serum at 39 °C and 5% CO2.

Transient transfections for DT40 in LacO array background were conducted as previously described using GFP-fused LacI, LacI:RM$^{WT}$, LacI:RM$^{RAXA}$ LacI:KI67$^{PP1BD}$ constructs[9,26].

For RNAi treatments, HeLa cells in exponential growth were seeded in six-well plates containing glass coverslips and grown overnight. Transfections were performed using Polyplus jetPRIME (PEQLAB, Southampton, UK) with the indicated siRNA oligos and analysed at 48 h after transfection as previously described[9]. For the rescue experiments HeLa cells at 50% confluence were transfected with 400 ng of plasmid DNA and 50 nM of siRNA oligonucleotides and analysed 48 h post-transfection. The siRNA oligonucleotides against Repo-Man (CDCA2) and SDS22 (PPP1R7) were obtained from Qiagen, Hs_CDCA2_5 and PPP1R7_7, respectively, Nup153_1 5′-GGCAGACU CUACCAAAUGUtt-3′ and Nup153_2 5′-GGACUUGUUAGAUCUAGUUtt-3′, and finally CGUACGCGGAAUACUUCGAdTdT was used as a control.

For rescue experiments, as in Fig. 3a, the oligo 5′-UGACAGACUUGACC AGAAATT-3′ was used instead of Hs_CDCA2_5.

Constructs and cell lines used in this study, were generated in Vagnarelli et al.[9]; HP1:GFP and Nup153:GFP construct were a kind gifts from Schirmer Lab (Welcome Trust Edinburgh) and Ullman Lab (University of Utah), respectively.

**Immunofluorescence microscopy.** Cells were fixed in 4% PFA and processed as previously described[26]. Primary antibodies were used as in Supplementary Table 1. Fluorescence-labelled secondary antibodies were applied at 1:200 (Jackson ImmunoResearch). Three-dimensional data sets were acquired using a wide-field microscope (NIKON Ti-E super research Live Cell imaging system) with a numerical aperture (NA) 1.45 Plan Apochromat lens. The data sets were deconvolved with NIS Elements AR analysis software (NIKON). Three-dimensional data sets were converted to Maximum Projection in the NIS software, exported as TIFF files, and imported into Adobe Photoshop for final presentation.

Live cell imaging was performed with a Nikon Ti-E super research live cell imaging system microscope as previously described[9].

For quantification of the staining in RNAi background masks were created around the DAPI nuclei. Mean intensity of antibodies signals were extracted and background was subtracted. For quantification of enrichment in LacI/LacO systems, three circles were designed around the LacI spot, within the nucleus and outside the cell and signals intensities were extracted. The outside circle served as background and was subtracted from both the nuclear and the LacI spot, then the signal intensity from the LacI was normalized relative to the intensity of the nuclear signal.

**PLA.** Proximity ligation assay was performed according to the manufacturer's protocol (Sigma). HeLa cells were fixed, permeabilized and blocked with BSA as previously described[9]. The antibodies were used at a concentration as per Supplementary Table 1. PLA probes were added and ligation was performed following manufacturer instructions (Sigma). Coverslips were mounted on DAPI and observed on the previously mentioned wide-field NIKON microscope. Spots lying within nuclear masks were counted in control and Repo-Man siRNA experiments.

**Quantitative real-time PCR.** RNA was collected from RNAi treated HeLa cells and extracted using the Tissue and Cells RNA Isolation Kit (Mobio) according to manufacturer's protocol. One microgram of RNA was used to prepare cDNA using the cDNA synthesis kit (Thermo Scientific) and oligo(dT) primers according to manufacturer's instructions. qPCR was quantified using SYBR green Master Mix (Thermo Scientific) and according to manufacturer's instructions (primers in Supplementary Table 2). Delta–delta CT method was used with normalization for GAPDH.

**Immunoelectron microscopy.** Cells were fixed in $2 \times 4\%$ paraformaldehyde, 0.2% glutaraldehyde in PHEM buffer (60 mM PIPES, 25 mM HEPES, 2 mM MgCl$_2$, 10 mM EGTA, pH6.9) for 60 min. Then $1 \times$ fix was added without the glutaraldehyde. Cells were scraped off the culture dish, pelleted, stored 1–2 days, resuspended in 15% PVP, 1.7 M sucrose in 0.1 M phosphate buffer with 33 mM Na$_2$CO$_3$, pH 7.4 and frozen in liquid nitrogen. Frozen pellets were sectioned on a cryo-ultramicrotome (Leica, UC6 with FC6 cryo-attachment). Cryosections, retrieved in 15% PVP, 1.7 M sucrose, were thawed, rinsed in PBS with 1% glycine, incubated in PBS with 1% BSA, incubated with rabbit anti-GFP antibody (Abcam) at 1:100 dilution, rinsed in PBS then incubated with the secondary anti-rabbit 10 nm colloidal gold (BBI Solutions). Grids were then rinsed in PBS, transferred to 1% glutaraldehyde (Agar Scientific) in PBS, washed in water and embedded in 2% methyl cellulose containing 0.4% uranyl acetate (Agar Scientific). Images were taken on a Hitachi H7600 electron microscope at 100 kV. For quantification of gold labelling, 50 images were acquired at a magnification of 60,000 times, corresponding to 5 μm$^2$ of cell area. Analysis was carried out using Fiji. The Freehand selection tool was used to measure the total nuclear area within the images analysed, delineated by the inner nuclear membrane or the edge of the image, as well as to estimate the area of the peripheral heterochromatin. Distinct nuclear bodies were only analysed if they were labelled.

**In vitro binding array.** The peptide array was purchased from Active Motif. GST: Repo-Man$^{403-1023}$ GST: Repo-Man$^{1-135}$ (ref. 9) or GST alone was expressed in Rosetta and purified on glutathione beads (Thermo Scientific). One micromolar of protein was processed onto the histone peptide array using the anti-GST (Pierce CAB4169, 1:1,000) and c-myc (positive control) as described by the manufacturer (Active Motif). LiCor secondary antibodies (LiCor IRDye 800CW and 680RD at 1:3,000 dilution) were used to allow imaging with the Odyssey system. Arrays were analysed through manufacturer's software.

**Preparation of Repo-Man bound nucleosomes.** Chromatin was extracted from HeLa cells and digested with Micrococcal Nuclease (NEB, 37 °C, 20 min) and incubated overnight (4 °C) with either GST:Repo-Man or GST alone

glutathione beads (Thermo Scientific) in binding buffer (50 mM TRIS, 1 mM CaCl$_2$, 4 mM MgCl$_2$, 0.32 M sucrose, 150 mM NaCl and 0.1% NP-40). Bound fraction was washed with binding buffer and eluted with glutathione reduced. DNA extracted and sequenced in Illumina HiSeq2500.

**Protein assays and quantitative immunoblotting.** HeLa cells were pelleted and prepared for blotting either through sonication in SDS sample buffer or fractionated according to the Subcellular Protein Fractionation Kit (Thermo Scientific). Membranes were incubated with primary antibodies as in Supplementary Table 1 and subsequently with IRDye-labelled secondary antibodies (LiCor). Fluorescence intensities were determined using an LiCor Odyssey CCD scanner according to manufacturer's instructions (LiCor Biosciences).

**Mass spectrometry.** Part of the chromatin eluted from the GST:Repo-Man or GST alone was loaded on an SDS–PAGE then stained with Instant Blue (Expedeon). The regions of gel containing the histones were excised and sent for Mass spectrometry.

Excised gel bands were de-stained and proteins were digested with trypsin, as previously described[65]. In brief, proteins were reduced in 10 mM dithiothreitol (Sigma) for 30 min at 37 °C and alkylated in 55 mM iodoacetamide (Sigma) for 20 min at ambient temperature in the dark. They were then digested overnight at 37 °C with 12.5 ng μl-1 trypsin (Pierce).

MS-analyses were performed either on an LTQ-Orbitrap mass spectrometer (Thermo Scientific) or on a Q Exactive mass spectrometer (Thermo Scientific) both coupled on-line to Ultimate 3000 RSLCnano Systems (Dionex, Thermo Scientific).

The MaxQuant software platform version 1.5.1.2 was used to process the raw files and search was conducted against Homo sapiens complete/reference proteome set of UniProt database (released on 14/05/2014), using the Andromeda search engine. For the first search peptide tolerance was set to 20 p.p.m. while for the main search peptide tolerance was set to 4.5 pm. Isotope mass tolerance was 2 p.p.m. and maximum charge to 7. Digestion mode was set to specific with trypsin allowing maximum of two missed cleavages. Carbamidomethylation of cysteine was set as fixed modification. Oxidation of methionine, acetylation, single, di- and tri-methylation of lysine, as well as single and di-methylation of arginine were set as variable modifications. Peptide and protein identifications were filtered to 1% FDR.

Histone PTM were detected only amongst the three GST:Repo-Man$^{CTerm}$ data sets and none on GST alone. Histone variants peptide counts mentioned in the text (H2AZ) or represented in Fig. 5a were over-represented in the GST:Repo-Man$^{CTerm}$ (at least 3-fold) when compared with GST alone.

For the characterization of serine 28 phosphorylation during mitosis, cells were grown overnight with nocodazole and mitotic extracts were collected and ran on a gel and stained with Instant Blue (Expedeon). The histone bands were excised for Mass Spectrometry. For the determination of the degree of phosphorylation on H3S28 two similar histone gel bands were digested as previously described. Before the addition of trypsin one of the samples was treated with alkaline phosphatase for 30 min at 37 °C. The analyses of phosphorylated S28 peptides was conducted as described in Steen et al.[66]

**Bioinformatic analyses.** Sequencing libraries were constructed, quantified and analysed according to standard protocols. Sequencing libraries were constructed on the Apollo 324 Next Generation Sample Preparation system (Wafergen) using the PrepX Complete ILMN 32i DNA Library Kit (Wafergen) according to the manufacturer's guidelines. The prepared libraries were quantified and multiplexed before 50-nt paired end sequencing on a HiSeq2500 (Rapid mode) according to standard Illumina protocols. Approximately 60–80 million read pairs were produced per sample and mapped to the human reference genome (hs37d5 version of build 37). Bam files from individual sequencing lanes were merged using Picard (Picard, http://broad-institute.github.io/picard/). Mapped reads were analysed for standard ChIP-Seq quality metrics; in particular, for each sample, the Normalized Strand Cross-correlation was > 1.05 and the Relative Strand Cross-correlation coefficients was > 0.8, suggesting a good degree of enrichment for the protein of interest, in agreement with[67]. Peaks were called using the software MACS2 with default parameters for narrow regions. Peaks located on unlocalized genomic contigs (for example, GL000192.1 or hs37d5) were excluded from the final set of significantly enriched regions. The sequences obtained with the GST alone were subtracted from the data sets.

Using a 5% FDR cut-off, 7550 binding sites were detected from the first and 4201 from the second duplicate. A stringent approach was applied to select 634 Repo-Man binding sites resulting from the union of common sites found in replicates 1 and 2.

HeLa broad peaks of histone markers and chromatin proteins of interest were downloaded from the ENCODE depository (https://genome.ucsc.edu/ENCODE/) and compared with Repo-Man-binding sites in terms of overlapping peaks using a Python script. UCSC Genome browser was used to visualize Repo-Man and ENCODE data sets.

Characterization of Repo-Man binding sites was performed using ChromHMM states downloaded from the Roadmap Epigenomics Database for the HeLa epigenome and using a 15-state Hidden Markov Model (HMM)[43].

Subtelomeric regions were defined as 500-kb windows adjacent to the terminal fragment of each chromosome as in Yang et al.[68]

**Chromosome positioning.** HT1080 cell line carrying a LacO integration on chromosome 13q22 and expressing a LacI:GFP (kindly provided by W Bickmore) was used with RNAi for Repo-Man as described before. Images of LacI:GFP co-stained with Nup153 were taken and analysed with the nuclear erosion scrip[69] to assess chromosome 13q22 location in relation to each of the five concentric shells.

**3D FISH.** The PAC CTC-820M16, localized in the subtelomeric region of chromosome 14 (14q32.33, Chr14:107106019-107206128 Ensemble draft 75 (ref. 45)) was labelled by nick translation with digoxigenin –dUTP (Roche), using the Abbott Molecular Nick Translation kit, as per manufacturer instructions. The 22-14 alpha satellite probe p14.1 (ref. 70) was similarly labelled with biotin-dUTP.

For 3D FISH, the transfected cells were incubated in CSK buffer (0.1 M NaCl; 0.3 M Sucrose; 0.003 M $MgCl_2$; 0.01 M Pipes) for 10 min, and then fixed in 2% formaldehyde/1 × PBS for 5 min. Cells permeabilization was carried out in 0.5% Triton X-100/1 × PBS for 20 min. Following an incubation in 0.1 N HCl for 10 min, and a wash in 2 × SCC, the probes were applied onto the cells, and the probe and nuclear DNA were denatured simultaneously at 85 °C for 5 min. The slides were incubated at 37 °C. The following day, slides were washed three times in 0.1 × SSC at 65 °C.

The probes were detected with anti-digoxigenin antibody, conjugated with rhodamine (Roche), or avidin Alexa Fluor 488 Conjugate (Invitrogen), both at 5 µg ml$^{-1}$, and the slides mounted in Vectashield DAPI (Oncor). Images were acquired with an Olympus BX-51 epifluorescence microscope coupled to a JAI CVM4+ CCD camera, with Leica Cytovision Genus v7.1.

**Chromatin immunoprecipitation.** ChIP was performed using the ChIP-IT express kit (Active Motif) according to Manual's Instructions. The protocol was erformed on RNAi treated HeLa cells and using 10 µl of H3K27me2/3 antibody (Active Motif, 39535) and digoxin as a negative control (Jackson Laboratories, 200-002-156). After ChIP, DNA was purified using Phenol Chloroform and ethanol precipitation with glycogen. DNA concentration was measured using Qubit (High Sensitivity Kit, Thermo) and reduced to 0.2 ng µl. qPCR was quantified using SYBR green Master Mix (Thermo Scientific) and according to manufacturer's instructions (primers in Supplementary Table 2). Delta–delta CT method was used with normalization for Input DNA. MYT1 gene was used as a control for H3K27me2/3 enrichment.

**Data availability.** Mass Spec data generated in this study have been deposited in PRIDE under accession number PXD004613. Hi-Seq data generated in this study have been deposited on Gene Expression Omnibus under GEO accession number GSE84035. The microscopy data are available from the corresponding author upon request and will be released via figshare.

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

## Acknowledgements

We thank Bridger's and Sala's groups (Brunel University London) for reagents and discussions, Dr Inês de Santiago (Cancer Research UK) for advice on bioinformatics analysis. We thank Dr Faidra Partheniou (Cytocell Ltd) for providing CTC-820M16 genomic location and Dr Priya Karan (Active Motif) for advice. This work was mainly supported by the BBSRC grant BB/K017632/1 to P.V. (IC, LL). M.L.d.G. was supported by a Placement ERASMUS fellowship. The work of the Wellcome Trust Centre in Oxford and the High-Throughput Genomics Group (for the generation of sequencing data) at the Wellcome Trust are supported by the Wellcome Trust 090532/Z/09/Z. J.I.d.H. and E.C.S. are supported by Wellcome Trust grants 095209 to E.C.S. and 092076 for the Centre for Cell Biology. M.W.G. is supported by the BBSRC grant BB/G011818/1. W.A.B. is supported by a University Unit grant from the Medical Research Council UK.

## Author contributions

I.J.d.C. and P.V. designed, performed and analysed the experiments. J.B., M.L.d.G., D.M., C.R. and M.W.G. performed and analysed experiments. L.L. and E.G. contributed with unpublished constructs and technical assistance. C.S. and J.R. performed and analysed mass spectrometry data sets. J.I.d.l.H. and V.V. performed analysis of sequencing data sets. S.S. and S.L. generated sequencing libraries and corresponding data files. E.C.S., K.S.U., W.A.B., C.G. and J.R. contributed with reagents, intellectual input and critical reading of the manuscript. I.J.d.C. and P.V. wrote the manuscript.

## Additional information

**Competing financial interests:** The authors declare no competing financial interests.

