## [Peer Review File · Nature Communications]

Reviewers' comments:

Reviewer #1 (Remarks to the Author):

Repo-Man is a nuclear protein that has been described as a chromosome-binding subunit of protein phosphatase 1 in mitosis and a regulator of DNA damage in interphase. This manuscript shows data to indicate that Repo-Man/PP1 also has an important function in heterochromatin organization at the nuclear periphery in interphase. The work is interesting but at least some data are preliminary or not conclusive. Direct evidence that Nup153 maintains endogenous Repo-Man at the nuclear pores in interphase is missing. The relevant histone docking site(s) for Repo-Man at the nuclear pores (histone H2B, H3 and/or histone H4?) remain(s) to be identified. It is difficult to grasp how distinct modifications of H3 and H4 (methylation and acetylation) both enhance the binding of Repo-Man, and how the knockdown and overexpression of RepoMan can have the same effect on HP1 binding. The link between H3S28 dephosphorylation by Repo-Man/PP1, which is only demonstrated in mitosis, and enhanced HP1 binding to H3K9me3 in interphase, is speculative. Finally, the manuscript is difficult to read because of a lack of key information in the legends.

Specific comments

1. There are numerous problems with the figures and the legends. Many figure panels are drawn with a powerpoint(-like) program and not of publication quality. The labeling of panels with numbers, as such or with ' or ', is confusing and hampers reading. A labeling with letters seems more appropriate. The figure panels are not always described in the order of their presentation, which also interferes with the reading.
2. p2, para 2: LADs is not defined. p5, para 2: it is not correct that this is the first study to show a role for PP1 complexes in epigenetic regulation. The introduction should also refer to a study indicating that Repo-Man has a role in DNA repair in interphase.
3. Fig. 1A lacks PP2A as a Repo-Man binding protein. There is no conclusive evidence that histone H2B binds directly to Repo-Man.
4. Fig. 1B: it is not indicated what is shown in the left two panels. It is not clear from the presented data whether Repo-Man interacts directly or indirectly with Nup153. Mapping of Nup153-Repo-Man interaction sites would possibly enable the generation of binding mutants that could be used to substantiate the currently unjustified claim that 'Nup153 acts as a platform to enrich or maintain Repo-Man at the nuclear periphery' (p7, last sentence of para 1).
5. Fig. 1C: The PLA staining is not convincing as most dots appear to be cytoplasmic. It is not clear what is shown in panel 3: is this with endogenous or ectopically expressed Nup153? The lower right figure seems to be empty; if so, it should be white.
6. Fig. 1D: panel 3 is not explained in the legend.
7. Fig. 1E lacks statistics.
8. Fig. 1F is not really convincing: the white arrows are not defined. The edges of the NPC are difficult to discern, which makes it impossible to draw conclusions about the exact localization of Repo-Man. A control staining with GFP alone is missing. A size bar is missing.
9. Fig. 1G: it is not clear how the indicated areas were defined. Also, the enrichment of Repo-Man at

intranuclear bodies is about 10-fold higher (gold particles/ square micrometer) than at the NPCs, which raises questions about the relevance of the interactions at the NPC.

10. Fig. 2A: a co-staining of Nup153 with endogenous Repo-Man is missing.

11. Fig. 2B: it is not clear what the two panels show.

12. Fig. 2C: the added value of this figure can be questioned.

13. Fig. 2D: the difference between the white and green staining is not explained.

14. Fig. 2E: it is not clear how the nuclear periphery was defined. The meaning of the numbers in the x-axis (1,2) were not explained.

15. Fig. 3A. Many error bars and the statistics are missing.

16. Fig. 3B. The inset figure is not explained. The RAXA mutant reduces but does not abolish HP1 recruitment. It is not shown that the Ki67 phosphatase-binding domain recruits PP1 as predicted.

17. Fig. 3C: it is not indicated which type of correlation coefficient is indicated. The same applies to panel 3H.

18. Fig. 3H: it is not clear why a correlation with K9 acetylation was shown here since HP1 binds to methylated H3K9.

19. Fig. 4A: it is difficult to understand how both acetylation and methylation of H3K27 enhance the binding of histone H3 to Repo-Man since they are chemically very different modifications. Also, they are markers for active and inactive chromatin, respectively, which makes it hard to explain why Repo-Man is enriched at heterochromatin. Methylation of histone H4 at K20 is equally efficient in enhancing the binding to Repo-Man but this is not further explored. In general, the data on histone binding would benefit a lot from the use of a histone-binding mutant of Repo-Man.

20. Fig. 4B. The legend says that Repo-Man has less affinity for peptides containing modified K27 residues. Actually, the opposite is suggested by the data.

21. Fig. 4C: the numbers (salt concentration?) are not explained in the legend. The interaction of Repo-Man with histones is apparently not very strong since it is abolished with nearly physiological salt concentrations. This also suggests that the interaction is of an ionic nature, which makes it even more difficult to understand why mono-methylation (no effect on charge) of histones increases their interaction with Repo-Man.

22. Fig. 4d: PLA dots do not show a co-localization with the nuclear periphery where Repo-Man is supposed to organize interphase chromatin (see comments on Fig. 1C).

23. Fig. 5A: the actual data should be shown

24. Figs. 5C and D: the abbreviations are not explained. A statistical analysis is missing.

25. Fig. 5E cannot be understood from the information in the legend.

26. Fig. 6B and D: the meaning of the numbers on the x-axis is not explained. Statistics are missing in

D. It is not clear what the statistics refer to in panel B.

27. Fig. 6E: it is not clear what the y-axis actually shows. Statistics are lacking.

28. Fig 7A-C: the data indicate that Repo-Man/PP1 dephosphorylates histone H3S28 in mitosis. What is needed here is evidence that this phosphatase dephosphorylates H3S10 and H3S28 in interphase.

29. Fig. 7D: trimethylation of H3K27 by EZH2 (PRC2) creates a docking site for PRC1, not for PRC2. The role of histone H4 modification is missing in the scheme. The proposed link between dephosphorylation of H3S28 and HP1 binding is not indicated.

30. It is speculated in the discussion section that only a fraction of H3S28 is phosphorylated during mitosis but the evidence/reference is lacking.

31. Supplementary illustrations:

- Fig.2A: the two experimental conditions should show the same time points. Fig. 2C: the color code should be swapped. The finding that both the knockdown and overexpression of Repo-Man decrease the number of HP1 foci is worrisome and not explained. Fig. 2D cannot be understood with the available legend.

- Figs. 3A and B, and Fig. 4B are not mentioned anywhere.

- There appear to be no legends to the supplementary tables.

Reviewer #2 (Remarks to the Author):

This manuscript demonstrates that Repo-Man/PP1 is a major phosphatase that modulates a repressive chromatin state.

This group has previously shown that Repo-Man interacts with Nup153, and the current study shows that enrichment of Repo-Man at nuclear periphery is Nup153-dependent. The authors further performed the loss-of-function assay and the tethering-recruiting experiment to address that Repo-Man/PP1 complex is necessary and sufficient to sustain the level of repressive histone marks H3K27me2/3 and H3K9me3, as well as the HP1 recruitment in the interphase nuclei. In addition, they showed that Repo-Man/PP1 is responsible for dephosphorylation of H3S28 in anaphase and cytokinesis. Intriguingly, presence of the H3S28ph abolishes Repo-Man binding to H3K27me2/3. Therefore, they propose a novel model that Repo-Man/PP1 modulates the switch of phospho/methyl at the H3 tail to establish a repressive chromatin environment after mitosis.

Overall, the data are convincing to support the role for Repo-Man in heterochromatin formation, and the manuscript is clearly written.

Specific comments:

1. The authors try to implement the idea that Nup153 tethers Repo-Man at nuclear pore basket to maintain repressive state of chromatin at the nuclear periphery (Fig. 7d and page 19, paragraph 2). However, there is no direct evidence linking the nuclear periphery-localized Repo-Man to chromatin organization. The model can be tested by checking whether the chromosome loci (13q22 or subtelomeric region of Chr14) move toward inside of the nucleus in cells where Repo-Man no longer stays at the nuclear periphery (such as upon Nup153 RNAi). It can also be checked if the gene expression of the selected telomeric genes (Fig. 6e) increases upon Nup153 RNAi.

2. Upon Repo-Man RNAi, some cells had the chromosome loci of 13q22 or chr14 subtelomeric region associate with the nuclear periphery (fig. 6a-d). Does delocalization of these chromosome loci directly correlate with a decreased level of H3K27me2/3 or H3K9me3 on these chromatin regions?
3. Page 3, the first sentence of the second paragraph should cite appropriate references.
4. Figure 6c, d: Description of the shell 1, 2+3, and 4+5 should be made clear.
5. Page 17, lane 3: H2S28 should be H3S28
6. Page 17, lane 10-12 and Fig. 7a: there is no data for H3T3.

Reviewer #3 (Remarks to the Author):

Repo-Man is a chromatin-binding PP1-targeting protein with well-characterized roles in dephosphorylating histones during and particularly after mitosis. Here, De Castro et al. investigate possible roles of Repo-Man in interphase. They argue that Repo-Man associates with Nup153 at the nuclear periphery, that it is found most prominently at sub-telomeric regions of chromosomes, and that it regulates telomere association with the nuclear periphery and the formation of repressive chromatin states. They imply, but do not show directly, that this regulation occurs because Repo-Man-PP1 dephosphorylates histones in interphase. The study addresses a fundamentally important process: the re-formation of heterochromatin foci following cell division. The findings in the paper are novel and, if substantiated, certainly interesting for those in the fields of histone modifications, phosphatases, mitosis, heterochromatin and telomere biology. The experiments cover multiple aspects of Repo-Man function, but don't quite nail down some of these definitively. Also, covering so many features of Repo-Man makes the logical flow of the paper a little difficult to follow in places. With confirmation of some key results, I think this would be a very interesting addition to the literature.

Main points

1. Although the story starts with a possible interaction between Nup153 and Repo-Man, these data are not completely compelling, and it is not clear to me that they are needed to support the idea that Repo-Man regulates heterochromatin. Although Repo-Man does appear somewhat enriched at the nuclear periphery, (i) this is not the major localization of Repo-Man (by far the highest density is in "intra-nuclear bodies" (Figure 1g), (ii) this is mainly shown for Repo-Man N-terminus, not for full-length protein (Figure 1f,g); (iii) Repo-Man signals in EM often seem far away from the nuclear membrane/NPC (Figure S1); (iv) there is no protein-protein interaction data to support direct binding; and (v) the effect of Nup153 RNAi on Repo-Man location could be indirect. Of note, because Nup153 is a NPC protein, it is an expected result that PLA between Nup153 and Repo-Man antibodies will predominantly be found in the nuclear periphery, even if only a very small proportion of Repo-Man antibody reactivity is in proximity to Nup153. Does Repo-Man RNAi reduce the number of PLA spots in experiments such as Figure 1c-e?
2. Related to point 1, the effect of Repo-Man on telomere location might not have much to do with Nup153 binding. Does Nup153 RNAi change telomere location using the assays of Figure 6?
3. The "phospho-methyl" switch has been used to refer to the case where S10P displaces reader

proteins from K9me2/me3, without reducing K9 methylation itself. In this study it is argued that S10P (and S28P) may regulate the methylation status of K9 (and K27), which is a different model. I think the distinction should be made clear. There is precedent for this latter model in some organisms (*Neurospora*, *Drosophila*) but this is perhaps the first time that evidence for this mechanism has been provided in human cells.

4. The model that Repo-Man-PP1 dephosphorylates S10P and S28P to allow heterochromatin formation predicts that S10P/S28P will increase at relevant loci when Repo-Man is depleted. Can this be tested, for example by IF or ChIPing S10P/S28P at telomeric or (normally) heterochromatic loci?

5. On a molecular level, it is surprising that Repo-Man could have affinity for such disparate modifications as H3K27me2, H3K27ac and H4K20me1. I am not aware of other histone readers that have this type of activity. The binding results seem very noisy in Fig S4A. Was the binding really reproducible in the two array experiments? Also, the band intensity in the unmodified H3 experiment (UB; Figure 4c3) is too weak to draw conclusions about the extent of the "Bound" fraction in this important control.

6. If Repo-Man is binding to H3K27me2/3, how can PLA between Repo-Man and H3K27me2/3 be detected? One would expect either that the anti-K27me2/3 antibody would displace Repo-Man, or that Repo-Man would prevent access to the methylation site by the anti-K27me2/3 antibody.

7. Some description of how the peptides identified by mass spec were analyzed to conclude that certain histones and histone modifications were significantly enriched should be provided. This is difficult to determine given the raw data provided in Table S3. Also, how was the GST alone control pull down data utilized here?

8. The finding that 70% of Repo-Man signals are found in RefSeq genes is very striking and must be significant. However, for all the findings in Fig 5c and 5d, it seems vital to know the observed/expected ratios as each of these features occupy different percentages of the genome.

Minor points

1. The method of binning into "shells" in Figure 6 needs to be outlined in the paper so it can be understood without reference to another study.

2. I do not see the H3T3 phosphorylation data that are mentioned in the text (Figure 7A).

3. The diagrams in Figure 7d and S6b are not very clear. It is difficult to see what the words refer to; it is not always clear what the arrows mean or what they are pointing to; and the significance of the H3K27me2/3 nucleosome "floating" in space is mysterious.

Reviewer #4 (Remarks to the Author):

As requested by the editor, I focused my review on the proteomics data presented in the manuscript. To begin, the proteomics data plays only a minor role in the overall manuscript, which uses a diverse array of techniques to characterize the involvement of Repo-Man in heterochromatin function. The authors carried out two replicate analyses of GST-Repo-Man to isolate HeLa nucleosomes and they searched for enrichment of certain post translational modifications. The data is provided in

supplemental table 3 and very briefly described on page 13. There are issues with the data and the presentation that need to be addressed by the authors. The authors should follow the the Molecular and Cellular Proteomics guidelines (<http://www.mcponline.org/site/misc/MSDataResources.xhtml>) for the description of the experiment and the presentation of the data. Also, authors must make their raw data available on a publically available server like Massive (<http://massive.ucsd.edu/ProteoSAFe/static/massive.jsp>). In addition, each of the supplemental tables, including supplemental table 3, need a detailed legend and description of the content. The only conclusions the authors seem to make using the proteomics data seems to be on page 13 where they state that the two independent repeats show preferential binding of Repo-Man to H2Az and H3.2 or H3.3, but they only list supplemental table 3 as the place to find this data. From supplemental table 3 there is no easy way to determine if this conclusion is supported by the data or not. A Figure summarizing the quantitative data to support this conclusion needs to be included in the manuscript or as a supplemental figure. Supplemental table 3 primarily lists lysine modified peptides, so it is hard to determine if their conclusion is correct or supported by the data. Also, given the fact that the modification data is available in in supplemental table 3, does the enrichment of histone modifications from the peptide array analysis correlated shown in Figure 4 a-b correlate with the proteomics analysis? This should be addressed in the manuscript. The other possibility for the authors is since the proteomics data is so minor in this manuscript and does not appear to make a significant contribution to the manuscript, it could be removed entirely from the work and probably not affect the conclusions of the work.

REVIEWERS' COMMENTS:

Reviewer #1 (Remarks to the Author):

The authors have gone to great length to constructively address the numerous comments that I raised in my first review report. They made considerable changes to the text and illustrations, and included added additional data. There are no further issues to be addressed.

Reviewer #2 (Remarks to the Author):

Revisions have greatly improved the manuscript.

One minor correction is necessary on page 21, line 490: H2S28 should read H3S28.

Reviewer #3 (Remarks to the Author):

De Castro et al. have made valiant attempts to address the reviewers' comments with significant new experiments (such as further Nup153 RNAi work). However, key elements supporting the model proposed remain unconvincing to my mind. In particular, it is still unclear how RepoMan can have affinity for such diverse modifications as H3K27me, H3K27ac and H4K20me. Great emphasis is placed on the H3K27me_{2/3} binding, and this is central to the model. However, there doesn't seem to be a particular reason to emphasize this over H3K27ac or H4K20me binding which, in the histone arrays, are actually stronger if anything. RepoMan binding to H3K27ac, for example, would be difficult to reconcile with a pro-heterochromatin action for RepoMan. Also, the data supporting RepoMan-H3K27me_{2/3} binding in cells is rather weak. First, there does not seem to be a good correlation between RepoMan and H3K27me_{2/3} and H3K9me₃ in the ChIP-seq data (and it is not clear why only correlations with selected dual modifications are assessed in the new version, not the single modifications). Second if, as argued by the authors, PLA between H3K27me_{2/3} and RepoMan involved RepoMan becoming linked to H3K27me_{2/3} on adjacent nucleosomes, then it cannot really be used as evidence of interaction as stated on line 318. While I appreciate that there may be technical difficulties, the failure to show that H3S28p is modulated at interphase heterochromatin foci is also a key missing element. Because an effect of RepoMan on heterochromatin formation was already published by these authors (Dev Cell 2011), I think a convincing molecular model for the underlying mechanism is required for publication in Nat Comms and this, unfortunately, is not quite provided by the current work.

Reviewer #4 (Remarks to the Author):

PRIDE is finally accessible and I have checked for the presence of the proper files on their website for this manuscript. The files are present and accessible, so the documentation of the proteomics data is in agreement with community standards.

All the datasets have been deposited according to the Journal requirement and can be accessed as below:

Mass Spec details of PRIDE submission for reviewers:

Project Name: Repo-Man/PP1 regulates heterochromatin formation in interphase

Project accession: PXD004613

Project DOI: Not applicable

Reviewer account details:

Username: reviewer24296@ebi.ac.uk

Password: oCfGsXNd

Hi-Seq Details of GEO submission for reviewers:

The following link has been created to allow review of record GSE84035 while it remains in private status:

<http://www.ncbi.nlm.nih.gov/geo/query/acc.cgi?token=kdetaiscddqfzg&acc=GSE84035>

Please see below the detailed response to how we have changed the manuscript addressing the referee comments.

Reviewer #1

We thank the referee for considering that “*The work is interesting*” and we have taken into account his/her suggestions to improve the manuscript and further strengthen our observations.

1. There are numerous problems with the figures and the legends. Many figure panels are drawn with a powerpoint(-like) program and not of publication quality. The labeling of panels with numbers, as such or with ' or ', is confusing and hampers reading. A labeling with letters seems more appropriate. The figure panels are not always described in the order of their presentation, which also interferes with the reading.

All the figures have been re-designed using Inkscape and saved as 600dpi. The labeling has been changed to letters and we have tried as much as possible to limit referring to figures in a different presentation order. In a few occasion it cannot be avoided since we have used the same kind of analyses and need to refer back to previous figures.

2 p2, para 2: LADs is not defined.

LADs are now defined and now reads “Lamina-Associated Domains (LADs)” within the text.

p5, para 2: it is not correct that this is the first study to show a role for PP1 complexes in epigenetic regulation.

The referee is correct and we acknowledge that we were not very clear stating this. We were referring to the first complex that acts on histones directly. NIPP1 has been clearly implicated in epigenetics but does not act directly as histone modifier. The sentence had been modified as “This represents the first study revealing Repo-Man/PP1 as an epigenetic regulator”.

The introduction should also refer to a study indicating that Repo-Man has a role in DNA repair in interphase.

We acknowledge that this is an important function for Repo-Man and the reference to Repo-Man involvement in DNA repair has been added: “while, in interphase is involved in DNA repair (Peng et al).”

3. Fig. 1A lacks PP2A as a Repo-Man binding protein. There is no conclusive evidence that histone H2B binds directly to Repo-Man

We have now added PP2A however the scheme refers to interactors in anaphase/interphase. PP2A binds only briefly at the anaphase onset and then is replaced by PP1. To differentiate the type of binding we have used a shaded color. In this graph we were not intending to show direct binding proteins but interactors.

4. Fig. 1B: it is not indicated what is shown in the left two panels.

The labels have been added to the panels.

It is not clear from the presented data whether Repo-Man interacts directly or indirectly with Nup153. Mapping of Nup153-Repo-Man interaction sites would possibly enable the generation of binding mutants that could be used to substantiate the currently unjustified claim that 'Nup153 acts as a platform to enrich or maintain Repo-Man at the nuclear periphery' (p7, last sentence of para 1).

We do not claim that the binding is direct. On the other hand, we provide compelling evidence that Nup153 is necessary for targeting Repo-Man (exogenous and endogenous) at the periphery. To strengthen this we have now added data on the effect of Nup153 RNAi on the enrichment of endogenous Repo-Man at the chromosome periphery. As discussed in the text this is the only possible approach to show endogenous Repo-Man at the periphery (Figure 2a and b). To complement this we have also added datasets (Figure 6 e, f and g) suggesting that depletion of Nup153 produces a phenotype similar to Repo-Man depletion in terms of loss of enrichment for heterochromatin marks at the periphery and gene expression changes.

Clearly the suggested sets of experiments are interesting and will be useful in the future as a follow up study.

5. Fig. 1C: The PLA staining is not convincing as most dots appear to be cytoplasmic.

I am not sure here what the referee refers to here. In the figure resented the majority of dots are within the DAPI staining (nucleus): 25 spots within the nucleus and 8 outside.

It is not clear what is shown in panel 3: is this with endogenous or ectopically expressed Nup153? The lower right figure seems to be empty; if so, it should be white.

We have taken into account that this needed clarification, as suggested by the referee. It is the PLA between the endogenous NUP153 and the endogenous Repo-Man. The text now reads like this “To understand this interaction at higher resolution, we first conducted proximity ligation assays (PLA) with antibodies against endogenous Repo-Man and Nup153”

We have also removed the black panel.

6. Fig. 1D: panel 3 is not explained in the legend.

The explanation for this panel has been added as follow: (d) HeLa cells were transfected with GFP:Nup153 (b, d, f) or GFP alone (a, c, e) and PLA (red) was performed using Repo-Man and GFP antibodies (c, d). Labels have been also added to the panels.

Note the change from number to letters as suggested by the referee in point 1.

7. Fig. 1E lacks statistics.

The statistics have been added

8. Fig. 1F is not really convincing: the white arrows are not defined. The edges of the NPC are difficult to discern, which makes it impossible to draw conclusions about the exact localization of Repo-Man. A control staining with GFP alone is missing. A size bar is missing.

We defined all the arrows and added a scale bar. Several other images showing the NPC localization are available in the supplementary figure 1 where now we have also added a representative image of GFP alone. The legend of the supplementary figure 1 has also been modified for clarity.

9. Fig. 1G: it is not clear how the indicated areas were defined.

A better description has been added to the Materials and methods. "For quantification of gold labelling, 50 images were acquired at a magnification of 60,000 times, corresponding to $5\mu\text{m}^2$ of cell area. Analysis was carried out using Fiji. The Freehand selection tool was used to measure the total nuclear area within the images analysed, delineated by the inner nuclear membrane or the edge of the image, as well as to estimate the area of the peripheral heterochromatin. Distinct nuclear bodies were only analysed if they were labelled."

Also, the enrichment of Repo-Man at intranuclear bodies is about 10-fold higher (gold particles/ square micrometer) than at the NPCs, which raises questions about the relevance of the interactions at the NPC.

This localization is actually not surprising since Repo-Man is also in the nuclear space and associated with heterochromatin, as discussed throughout the manuscript. Moreover, HP1 foci, for example, have been found across the nucleoplasm and here indeed we show that Repo-Man is involved in their maintenance.

The biological relevance of these intranuclear foci is not clear at the moment, although we agree it will be an interesting aspect to follow up. We have taken into consideration the points raised by the referee and clarified the text. The text now reads: "and a proportion of Repo-Man is also associated with intra-nuclear bodies in chromatin dense regions, which are possibly related to the proposed role of Repo-Man in heterochromatin formation (see later in the text); further studies however will be required to elucidate their nature (Fig. 1h, Supplementary Fig. 1)"

10. Fig. 2A: a co-staining of Nup153 with endogenous Repo-Man is missing.

As we have stated previously (point 4) we have now conducted the experiments suggested here by the referee and the staining is included in Figure 2 b.

11. Fig. 2B: it is not clear what the two panels show.

Fig 2B is now Figure 2d: The panels have been labeled.

12. *Fig. 2C: the added value of this figure can be questioned.*

(Now Figure 2e): We tried to better explain the figure in the legends, so the point of this figure would come across more clearly. Here we wanted to show how we have calculated the peripheral enrichment. For that, we have also included schematics in the figure per se as well as improved the legend – see point 14.

13. *Fig. 2D: the difference between the white and green staining is not explained.*

(Now Figure 2f): The panels have been labeled.

14. *Fig. 2E: it is not clear how the nuclear periphery was defined.*

(Now Figure 2g): we have explained these two points better in the text/legend: For the peripheral accumulation the Figure legend now reads as “Repo-Man enrichment was measured as the ratio between the average of the two maximum intensity values (Max1 and Max2) by the median of the values in the plateau (c and g).”

... *The meaning of the numbers in the x-axis (1,2) were not explained.*

We acknowledge this was not very clearly mentioned in the legend. Now the legend reads as follow: after RNAi with Control oligos (grey bars) or with a single (Nup153²) or combination (Nup153^{1&2}) Nup153 oligos (green bars).

15. *Fig. 3A. Many error bars and the statistics are missing.*

We added the significance levels in every figure and the analyses performed is indicated within the figure legend.

16. *Fig. 3B.*

The inset figure is not explained.

We have added the explanation in the figure legend: “Cells were fixed and stained with HP1 antibody (representative image shown in the inset).” and pointed to the Lacl spot with a green arrow in order to improve the understanding on this figure.

The RAXA mutant reduces but does not abolish HP1 recruitment.

The referee is correct: we have changed the text as follow: “HP1 recruitment is dependent on the phosphatase activity of the complex since it is significantly reduced by the Repo-Man RAXA mutant (PP1 non-binding mutant)”.

It is not shown that the Ki67 phosphatase-binding domain recruits PP1 as predicted.

We have shown this in a previous paper (referenced) using exactly the same system (Booth et al. eLife 2014;3:e01641. DOI: [10.7554/eLife.01641](https://doi.org/10.7554/eLife.01641): Figure 1 C-D).

17. *Fig. 3C: it is not indicated which type of correlation coefficient is indicated. The same applies to panel 3H.*

This has been specified in the figure legend for both panels.

18. *Fig. 3H: it is not clear why a correlation with K9 acetylation was shown here since HP1 binds to methylated H3K9.*

This is an important aspect and it is at the heart of the mechanism we are looking at here. HP1 detachment from H3K9me3 occurs when the S10 residue is phosphorylated. This can be a transient effect (as it occurs during mitosis) or long-term effect. While in the former situation the H3K9me3 remains on chromatin, in the

latter the methyl is converted to acetyl. (for a review on this see Priscilla Nga Leng Lau & Peter Cheung (2011) Unlocking polycomb silencing through histone H3 phosphorylation, Cell Cycle, 10:10, 1514-1515, DOI: 10.4161/cc.10.10.15433). We have extended our comments on this aspect in the discussion section in order to clarify the point raised by the referee.

19. Fig. 4A: it is difficult to understand how both acetylation and methylation of H3K27 enhance the binding of histone H3 to Repo-Man since they are chemically very different modifications. Also, they are markers for active and inactive chromatin, respectively, which makes it hard to explain why Repo-Man is enriched at heterochromatin. Methylation of histone H4 at K20 is equally efficient in enhancing the binding to Repo-Man but this is not further explored. In general, the data on histone binding would benefit a lot from the use of a histone-binding mutant of Repo-Man.

I understand the comment of the referee but these are the data. We have repeated the array analyses again with the GST:Repo-Man^{Cterm} which showed similar trend to the array that had been analyzed upon the first submission; the new figure combines all the datasets. As suggested by the referee we have conducted the experiment also with an histone binding mutant that still targets to the chromatin at the periphery in anaphase GST:Repo-Man¹⁻¹³⁵ (this mutant was described in Vagnarelli et al, Dev Cell 2011). The data have been added in a new Figure 4c. We have also tested the chromatin binding ability of the mutant compared to the wt. We now show that the GST:Repo-Man^{Cterm} but not the mutant (GST: Repo-Man¹⁻¹³⁵), binds chromatin in vitro (new figure 4d).

I do agree that these are weak interactions but a combination of weak interactions may confer a more stable binding. This latter aspect will need to be explored by chromatin reconstitutions experiments and will represent an important aspect for further studies.

We have added this point of clarification within the text.

20. Fig. 4B. The legend says that Repo-Man has less affinity for peptides containing modified K27 residues. Actually, the opposite is suggested by the data.

The legend was: "Repo-Man has less affinity for peptides containing modified K27 residue (either methylated or acetylated) and S28 Phosphorylation (S28P)" We have slightly changed the text to make it more explicit: "Repo-Man has less affinity for peptides containing modified K27 residue (either methylated or acetylated) if the adjacent S28 is Phosphorylated (S28P)"

21. Fig. 4C: the numbers (salt concentration?) are not explained in the legend. The interaction of Repo-Man with histones is apparently not very strong since it is abolished with nearly physiological salt concentrations. This also suggests that the interaction is of an ionic nature, which makes it even more difficult to understand why mono-methylation (no effect on charge) of histones increases their interaction with Repo-Man.

We have removed this figure in the new version of the manuscript since we could not obtain a clearer image for the unmodified peptide as requested by referee 2.

22. *Fig. 4d: PLA dots do not show a co-localization with the nuclear periphery where Repo-Man is supposed to organize interphase chromatin (see comments on Fig. 1C).*

To understand the point raised by the referee we have decided to add a panel showing the PLA signal together with the five concentric areas defined by the nuclear erosion script. This addition we believe will help the readers to understand this point: Figure 4e, clearly shows that the vast majority of spots are within the outer shells. The 2 external shells are conventionally classified as peripheral. We would also like to point out that Repo-Man is not only at the pores (as the EM images show).

23. *Fig. 5A: the actual data should be shown*

The data are uploaded in PRIDE and (PXD004613)

24. *Figs. 5C and D: the abbreviations are not explained. A statistical analysis is missing.*

(now Figure 5d and e) The statistical analyses are included for the modified version of Figure 5d and the figure legend reads as follow: “(d) Annotation of Repo-Man hits according to gene features or lamina associations ⁷ (Fisher p-values). TES: Transcription End Site; TSS: Transcription Start Site; LADs: Lamnia Associated Domains (e) Overlaps between Repo-Man hits and double histone modifications extracted from HeLa ENCODE datasets for H3K27ac, H3K4me3, CTCF, H3K9ac, H3K79me2, H3K27me3, H2AZ, H3K9me3, EZH2 and H4K20me1 (Fisher p-values).”

25. *Fig. 5E cannot be understood from the information in the legend.*

(now Figure 5f): The figure legend now reads as follow: (f) Single gene profiles of Repo-Man target genes *PPP2R2C* (1) and *PDE9A* (2) classified as polycomb repressed and heterochromatin associated (H3K9me3) respectively by the software ChromHMM ⁵³. The chromosomes and the position of the gene (red line) are shown along with the representation of the gene genomic sequence (lines/squares are exons). Repo-Man binding sites distribution is shown for two independent datasets (light and dark green). Positioning of histone marks along the genomic window were extracted from the UCSC in HeLa cells (H2AZ H3K9ac, H3K9me3, H3K27ac, H3K27me3, H4K20me1 and S2-PolII), reads in y axis = 50.

26. *Fig. 6B and D: the meaning of the numbers on the x-axis is not explained. Statistics are missing in D. It is not clear what the statistics refer to in panel B.*

We have changed the labeling of the X-axis in order to be more clear and have clarified the statistics in the legends.

27. *Fig. 6E: it is not clear what the y-axis actually shows. Statistics are lacking.*

(Now Figure 6g) The axes labeling has been changed to “Normalised Relative Expression (RM/Nup153 RNAi versus Ctrl RNAi)” and Statistics have been added.

28. *Fig 7A-C: the data indicate that Repo-Man/PP1 dephosphorylates histone H3S28 in mitosis. What is needed here is evidence that this phosphatase dephosphorylates H3S10 and H3S28 in interphase.*

The figures shown in 7a and b are of mitotic exit and the quantification of cells in anaphase or telophase/cytokinesis. We did not analyse here early mitosis.

Although we understand the importance of studying the removal of S28P in interphase, a completely different experimental set up needs to be used to this purpose. To stimulate the MSK1 cascade, it is required to starve cells and then stimulate them by several means. In HeLa cells, the system we have used in this paper, this was not possible despite several attempts and different strategies. We and others have tried but the cells go into apoptosis whether a low-serum or no-serum strategy is followed. To switch to a completely different cellular system, we need to map the Repo-Man binding site *de novo* in order to be able to correlate them to the effect seen.

Therefore, although very interesting, this is a big task that requires a detailed and extended study.

29. Fig. 7D: trimethylation of H3K27 by EZH2 (PRC2) creates a docking site for PRC1, not for PRC2. The role of histone H4 modification is missing in the scheme. The proposed link between dephosphorylation of H3S28 and HP1 binding is not indicated.

Here we have tried to clarify the scheme better. The scheme illustrates the role of Repo-Man in regulating the association with PRC2 through the regulation of S28 phosphorylation. We have modified the legend in order to clarify better the scheme.

Now the legend reads as: "Model. Repo-Man associates with modified Lysine 27 when the adjacent S28 is not modified. There, through the dephosphorylation of S28 residues of nearby nucleosomes, it regulates the association of PRC2 and maintenance of H3K27me; it also contributes towards HP1 associations by dephosphorylating S10. These processes allow the establishment and/or maintenance of a repressive environment"

We cannot add the role of H4 since we did not investigate this aspect.

30. It is speculated in the discussion section that only a fraction of H3S28 is phosphorylated during mitosis but the evidence/reference is lacking.

This knowledge came from several discussions with colleagues but we agree with the referee that a first proof for this is required in our system. To this purpose we have analysed the percentage of S28 residues that are phosphorylated in mitosis by mass spectrometry. This data has been added to supplementary Figure 6 d-e and the full dataset has been deposited in PRIDE.

The data obtained shows that 36.5% of S28 residues are phosphorylated in mitosis, at least in HeLa cells.

31. Supplementary illustrations:

- Fig.2A: the two experimental conditions should show the same time points.

We added the same time points but also we have shown an extra time point for the Repo-Man RNAi to illustrate that even at a later stage the foci are not reforming.

Fig. 2C: the color code should be swapped.

We have changed the color.

The finding that both the knockdown and overexpression of Repo-Man decrease the number of HP1 foci is worrisome and not explained.

Although we understand the reviewer concerns we disagree with the reviewer interpretation. We have added a more detailed explanation in the text in order to clarify our point of view. The text reads as: "From this picture it emerges that a local balance of active phosphatases is important to maintain the correct level of heterochromatin in cells. It is therefore expected that overexpression of these regulators, either by binding to non-canonical chromatin regions or titrating PP1 away from the bound targeting subunit, can produce an abnormal chromatin environment as well; this indeed appears to be the case since Repo-Man overexpression also contributes to disrupt the normal accumulation of HP1 foci in interphase nuclei (Supplementary Fig. 2c, d)."

Fig. 2D cannot be understood with the available legend.

We have changed the legend

Figs. 3A and B, and Fig. 4B are not mentioned anywhere.

We have taken care that all the figures are now mentioned.

There appear to be no legends to the supplementary tables.

All tables have the relative title

Reviewer #2 (Remarks to the Author):

We thank the reviewer for considering that "*the data are convincing to support the role for Repo-Man in heterochromatin formation, and the manuscript is clearly written*".

1.....The model can be tested by checking whether the chromosome loci (13q22 or subtelomeric region of Chr14) move toward inside of the nucleus in cells where Repo-Man no longer stays at the nuclear periphery (such as upon Nup153 RNAi). It can also be checked if the gene expression of the selected telomeric genes (Fig. 6e) increases upon Nup153 RNAi.

We thank the referee for this suggestion. We have tested, as suggested, the gene expression of the selected telomeric genes (Fig. 6e) upon Nup153 RNAi. As shown in a new graph in Figure 6g, the expression of the selected genes increases after the knock down of Nup153 as postulated by the referee.

2. Upon Repo-Man RNAi, some cells had the chromosome loci of 13q22 or chr14 subtelomeric region associate with the nuclear periphery (fig. 6a-d). Does delocalization of these chromosome loci directly correlate with a decreased level of H3K27me2/3 or H3K9me3 on these chromatin regions?

This was again a very insightful suggestion. We have in fact checked the level of H3K27me2/3 at some of the loci. Indeed the knock-down of Repo-Man leads to a statistically significant decrease of this mark at these loci. The data are now presented in Figure 5g and we believe it strengthened our conclusions.

3. Page 3, the first sentence of the second paragraph should cite appropriate references.

We added a review to cover the several important studies on this subject.

4. *Figure 6c, d: Description of the shell 1, 2+3, and 4+5 should be made clear.*

The annotations on the graphs and relative figure legends have been changed in order to give a more comprehensive understanding of the analysis.

5. *Page 17, lane 3: H2S28 should be H3S28*

This has been corrected.

6. *Page 17, lane 10-12 and Fig. 7a: there is no data for H3T3.*

The referee is correct: this was a mistake in the text. We have now corrected the sentence. The H3T3 data are part of a previous publication now referred to in the text.

Reviewer #3 (Remarks to the Author):

We thank the reviewer for the positive and constructive comments on our study and we hope that, thanks to the several new data added in the revised version, he/she would think that is now a “very interesting addition to the literature.”

1. *Although Repo-Man does appear somewhat enriched at the nuclear periphery, (i) this is not the major localization of Repo-Man (by far the highest density is in "intra-nuclear bodies" (Figure 1g),*

We have commented on this aspect in the text and see reply to Referee 1 – Point 9.

(ii) this is mainly shown for Repo-Man N-terminus, not for full-length protein (Figure 1f,g);

We understand the concerns raised by the referee. Unfortunately, there are no available antibodies that can detect the peripheral pool of Repo-Man in interphase, however a couple of antibodies (our own and the Abcam one) recognize this pool at the chromosome periphery during late anaphase/telophase. We have then used this system to address the question raised by the referee and the new data are presented in Figure 2b. As the referee can see the peripheral accumulation of Repo-Man is lost upon Nup153 RNAi as shown for the cell line (Figure 2a). Moreover the behavior of the full length or Repo-Man^{N^{Term}} in term of localization and dependency on Nup153 are identical (Figure 2 g).

(iii) Repo-Man signals in EM often seem far away from the nuclear membrane/NPC (Figure S1);

Supplementary Figure1 contains different nuclear localizations, NPC, peripheral heterochromatin and intra-nuclear heterochromatin bodies. To clarify better we have changed both the annotations on the figures and the figure legends. In addition, we have added a representative image of GFP alone labeling where the referee can see a widespread localization.

(iv) there is no protein-protein interaction data to support direct binding; and (v) the effect of Nup153 RNAi on Repo-Man location could be indirect.

We understand the concern raised by the referee but at the same time we do not claim that this is via a direct binding. Interactions and biological functions can be

mediated by indirect interactions but never-the-less being very important in establishing cellular processes. To clarify this aspect better we have now added onto the text: "Nup153 interacts (directly or indirectly) with Repo-Man"

Of note, because Nup153 is a NPC protein, it is an expected result that PLA between Nup153 and Repo-Man antibodies will predominantly be found in the nuclear periphery, even if only a very small proportion of Repo-Man antibody reactivity is in proximity to Nup153.

Yes, as pointed out by the referee Nup153 is mainly localized in the periphery. As for Repo-Man there is a pool associated at the periphery as seen by EM. However, both Repo-man (Figure 2c) and Nup153 are in the nucleoplasm as well, and recent work has shown that Nup153 binds to promoters of gene not necessary located at the nuclear periphery (Jacinto et al. 2015; Vaquerizas et al. 2010). Our results point that the interaction between the two proteins is mainly occurring in the peripheral compartment.

Vaquerizas, J. M. *et al.* Nuclear pore proteins nup153 and megator define transcriptionally active regions in the Drosophila genome. *PLoS Genet* **6**, e1000846, doi:10.1371/journal.pgen.1000846 (2010).

Jacinto, F. V., Benner, C. & Hetzer, M. W. The nucleoporin Nup153 regulates embryonic stem cell pluripotency through gene silencing. *Genes Dev* **29**, 1224-1238, doi:10.1101/gad.260919.115 (2015).

Does Repo-Man RNAi reduce the number of PLA spots in experiments such as Figure 1c-e?

We thought this was an interesting suggestion that would strengthen the results and we have conducted the suggested experiments and the results are now shown in Figure 1f.

2. Related to point 1, the effect of Repo-Man on telomere location might not have much to do with Nup153 binding. Does Nup153 RNAi change telomere location using the assays of Figure 6?

This is an interesting point also raised by Referee 2. Please see reply to referee 2 (point 1).

We have also added the analyses of the enrichment at the nuclear periphery for both H3K9me3 and H3K27me2/3 after Nup153 RNAi and the results are now added in Figure 6 e and f. These new datasets show that Nup153 RNAi leads to a loss of heterochromatin markers at the periphery and the expression across a panel of genes goes up.

3. The "phospho-methyl" switch has been used to refer to the case where S10P displaces reader proteins from K9me2/me3, without reducing K9 methylation itself. In this study it is argued that S10P (and S28P) may regulate the methylation status of K9 (and K27), which is a different model. I think the distinction should be made clear. There is precedent for this latter model in some organisms (Neurospora, Drosophila) but this is perhaps the first time that evidence for this mechanism has been provided in human cells.

We do agree it is an important point and we thank the referee to have added this insightful suggestion. We have expanded our discussion to better explain this point. You can now read "Multiple kinases, including RSK2, MSK1/2, PIM1, and IKK α , have been shown to directly phosphorylate H3 thereby indicating that H3

phosphorylation is a critical step in signal transduction to the chromatin/transcriptional regulatory machinery (reviewed in ⁷⁷). This has two possible implications: the presence of phosphorylation does not alter the remaining histone code but the association of chromatin readers; or the presence of phosphorylation has implications on nearby histone modifications. For example, Haspin phosphorylation of H3T3 has been implicated in the dissociation of TFIID from chromatin without altering the status of the nearby H3K4me3 ⁷⁸. Other example is that of the binding of polycomb proteins to K9me3, which is blocked by S10 phosphorylation, despite the fact that the level of K9me3 is maintained ⁷⁹. Moreover, stress induction of MSK1 can re-activate the polycomb-silenced α -globin gene via H3S28 phosphorylation ⁸⁰ and gene activation during ES cells differentiation through dissociation of PRC ⁸¹. The presence of H3S28P has also been associated with changes in the epigenetic landscape. A methyl/acetylation switch on the lysine 27 has been proposed in a luciferase reporter where MSK1 phosphorylation of S28P leads to K27 acetylation coupled with reduction of K27me3 and of polycomb group of proteins binding at the reporter ⁸⁰. Since we detect a decrease of H3K27me2/3 after Repo-Man RNAi at specific loci where Repo-Man is bound, we suggest that Repo-Man could be important in mediating the acetyl-methyl switch⁸⁰, through dephosphorylation of H3.”

4. The model that Repo-Man-PP1 dephosphorylates S10P and S28P to allow heterochromatin formation predicts that S10P/S28P will increase at relevant loci when Repo-Man is depleted. Can this be tested, for example by IF or ChIPing S10P/S28P at telomeric or (normally) heterochromatic loci?

The residue level of S28P in interphase is not sufficiently abundant to perform ChIP experiments in order to have a sufficient resolution power. We have tried to induce S28P in our system after starvation but without success. See also comment to Referee 1 Point-28.

5. On a molecular level, it is surprising that Repo-Man could have affinity for such disparate modifications as H3K27me2, H3K27ac and H4K20me1. I am not aware of other histone readers that have this type of activity. The binding results seem very noisy in Fig S4A.

Was the binding really reproducible in the two array experiments?

To address the concern of the Referee we have repeated the analyses with the array (Figure 4a): these latter experiments showed the same modifications detected in the array. The new and old datasets have now been combined in the new figure. In addition, as suggested by Referee 1 (point 19) we have also conducted the same experiment using a histone non-bind mutant, GST:RM¹⁻¹³⁵. The new updated quantifications and the new datasets have been added in Figure 4 a,b,c.

Also, the band intensity in the unmodified H3 experiment (UB; Figure 4c3) is too weak to draw conclusions about the extent of the "Bound" fraction in this important control.

We agree with the referee and, although after several repeats, we could not get better results for the peptide experiments. Therefore we have removed these data from the paper. However we have added an in vitro chromatin binding experiment that also includes the mutant form. These data are presented in Figure 4d (see comment above).

At a molecular level and as a point of speculation, we think that although the single histone interactions are weak interactions, the sum of all in the context of chromatin will produce a more robust binding as seen in vivo. Future experiment using in vitro chromatin reconstitution will be used to follow up on these findings.

6. If Repo-Man is binding to H3K27me2/3, how can PLA between Repo-Man and H3K27me2/3 be detected? One would expect either that the anti-K27me2/3 antibody would displace Repo-Man, or that Repo-Man would prevent access to the methylation site by the anti-K27me2/3 antibody.

This is an interesting point. I believe it happens in the same way in which we can see co-localisation between H3K9me3 and HP1; (HP1 binds the me3 of H3K9). However I think that the reason behind is because these regions are enriched for these modifications but their relative epigenetic readers bind only to a subset of the possible sites. Therefore the chromatin stretch will never be fully occupied either by HP1, PRC2 or Repo-Man. Therefore what we (and the rest of the field) really detect in these experiments (co-localisation or PLA) is the signal coming from one molecule in close proximity to sites highly enriched for these modifications. The other possible explanation is that we only see the dynamic state.

7. Some description of how the peptides identified by mass spec were analyzed to conclude that certain histones and histone modifications were significantly enriched should be provided. This is difficult to determine given the raw data provided in Table S3. Also, how was the GST alone control pull down data utilized here?

We agree this was not made clear before. All datasets generated here are deposited in PRIDE and an explanation on how the data was analysed is now provided in Materials and Methods.

8. The finding that 70% of Repo-Man signals are found in RefSeq genes is very striking and must be significant. However, for all the findings in Fig 5c and 5d, it seems vital to know the observed/expected ratios as each of these features occupy different percentages of the genome.

The referee here has a good point. We have added the statistical analyses of the datasets. The p values are displayed in the figures.

Minor points

1. The method of binning into "shells" in Figure 6 needs to be outlined in the paper so it can be understood without reference to another study.

We have added the details in the figure legend.

2. I do not see the H3T3 phosphorylation data that are mentioned in the text.

We apologise, it was a mistake. This has been corrected. Data on the H3T3 have already been published before and this is now stated in the text.

3. The diagrams in Figure 7d and S6b and not very clear. It is difficult to see what the words refer to; it is not always clear what the arrows mean or what they are pointing to; and the significance of the H3K27me2/3 nucleosome "floating" in space is mysterious.

We have tried to add some changes and to have a better legend to explain this.

Reviewer #4 (Remarks to the Author):

We apologise if the table was not of sufficient quality so we have removed the table and we have the complete datasets now deposited in PRIDE.

As a response to the specific editorial requests:

1) *Please also review the changes in the attached copy of your manuscript, which has been edited for style, and address the comments and queries I have added. Please note that these are high level comments for developmental editing and do not include copy edits. If using Word, please use the 'track changes' feature to make the process of accepting your manuscript more efficient.*

We have followed all the instructions given for editing, within the manuscript provided. Moreover the length and references are now in line with the requirement for Nature Communications articles.

2) *However, we would like a written response to these concerns in your next cover letter, adding additional text/modifying existing text to the main Discussion to include potential caveats about the enrichment of Repo-man at heterochromatin as needed. Referee 1 has told us that the definitive explanation re: binding of Repo-man to distinct Histone modifications is indeed lacking. As such, we would like the main text to tone down the conclusions, discuss shortcomings and potential alternative explanations. Formal re-review of the next version will not be necessary, but other editors and I will check the revised text to make sure that appropriate changes have been made.*

We have edited the text in several parts and the main changes in the statements and addressing potential alternative explanations are highlighted in blue in the accompanying manuscript.

We have added in the section below our response to referee 3 and we have modified the aspects of concerns and toned down the sentences suggested as follow:

The last sentence of the introduction

"Collectively our data shows that Repo-Man/PP1 chromatin interactions allow the transmission of a particular chromatin environment from one generation to the next, and by doing so, this complex contributes to shape nuclear chromatin structure and organisation in interphase.

This represents the first study revealing Repo-Man/PP1 as an epigenetic regulator that maintains a local fine-tuning of the histone code through histone phosphorylation switches"

Now reads: "Collectively, our data shows that Repo-Man/PP1 regulates the histone code and chromatin structure at least across a panel of target regions".

Please see below the detailed response to how we have changed the manuscript addressing the referee comments.

Reviewer #1 (Remarks to the Author):

The authors have gone to great length to constructively address the numerous comments that I raised in my first review report. They made considerable changes to the text and illustrations, and included added additional data. There are no further issues to be addressed.

We thank the Reviewer for his/her comments and we are pleased that we have addressed his/her previous criticisms.

Reviewer #2 (Remarks to the Author):

*Revisions have greatly improved the manuscript.
One minor correction is necessary on page 21, line 490: H2S28 should read H3S28.*

We thank the Reviewer and we have changed the highlighted mistake.

Reviewer #3 (Remarks to the Author):

De Castro et al. have made valiant attempts to address the reviewers' comments with significant new experiments (such as further Nup153 RNAi work).

We thank the reviewer for having recognized our efforts in addressing the criticisms raised by the referees.

However, key elements supporting the model proposed remain unconvincing to my mind. In particular, it is still unclear how RepoMan can have affinity for such diverse modifications as H3K27me, H3K27ac and H4K20me.

We are sorry if we failed to provide stronger evidence for this.

Maybe our text was not sufficiently clear in expressing our vision and presenting the different possibilities that can be advocated as explanations for our observations.

We have changed the text in several points to make it clearer (see highlighted blue text within the main manuscript).

We do recognize and acknowledge in the text that these modifications are quite different. However, as far as the biology of this complex stands today, we do not know if more than a single chromatin-targeting region lies within the large C-Term domain we used as bait. Therefore, there is the formal possibility that two separate domains can direct the protein towards different chromatin regions.

This consideration is supported by some unpublished results we have obtained and that we are happy to confidentially share with the editors:

The previously identified chromatin binding domain (regulated by Aurora B phosphorylation) is indeed important for Repo-Man targeting and we have confirmed that the phospho-mimetic mutant decreases the binding in vivo (figure below, panel c and in vitro (Supplementary figure 4E). However, our careful analyses also show

that it does not abolish the binding to chromatin both in vitro and in vivo (panel c with d – a LaminA construct that is clearly excluded from chromatin_ in the figure below). Moreover, a Repo-Man mutant (aa 1-900) which contains all the domains so far identified for targeting to chromatin, does not fully accumulate on chromosomes as the wild type. All these data suggest that there might be a more complex way of Repo-man targeting to chromatin than the one so far identified. We have revised the text including this latter as a possibility.

Great emphasis is placed on the H3K27me2/3 binding, and this is central to the model. However, there doesn't seem to be a particular reason to emphasize this over H3K27ac or H4K20me binding which, in the histone arrays, are actually stronger if anything. RepoMan binding to H3K27ac, for example, would be difficult to reconcile with a pro-heterochromatin action for RepoMan. Also, the data supporting RepoMan-H3K27me2/3 binding in cells is rather weak. First, there does not seem to be a good correlation between RepoMan and H3K27me2/3 and H3K9me3 in the ChIP-seq data (and it is not clear why only correlations with selected dual modifications are assessed in the new version, not the single modifications).

Again, in this respect, we have added some clarification in the text and changed some claims that could have been mis-leading.

Nevertheless, our mass spec datasets suggest that Repo-Man pulls down the K27me3 only when in conjunction with H3.3 variant and not within the H3.1 or H3.2. On the contrary, the H3K27ac is pull-down only when associated with H3.1 or H3.2 (Figure 5a).

These data indicate that Repo-Man binding to chromatin is not a simple and one-hit process and it is the reason behind our choice of presenting overlaps with more than

a single modification. Our interpretation, suggested by these experiments, is that Repo-Man recognizes a complex chromatin signature that possibly involves interactions with different histone modifications. In fact, each of these single interactions are quite weak interactions however, in cells, Repo-Man is quite tightly bound to chromatin as indicated by the harsh conditions required to dissociate it from chromatin and from the FRAP studies.

We do agree with the Referee that more studies, possibly involving in vitro chromatin reconstitutions, will be necessary to have a detailed picture of all the elements and relative weight of all the interactions that direct Repo-man binding.

Second if, as argued by the authors, PLA between H3K27me2/3 and RepoMan involved RepoMan becoming linked to H3K27me2/3 on adjacent nucleosomes, then it cannot really be used as evidence of interaction as stated on line 318.

We have changed the sentence as follow:

“This indicates that the Repo-Man is indeed enriched at chromatin regions containing H3K27me2/3 *in vivo* and in proximity of the nuclear periphery compartment”

While I appreciate that there may be technical difficulties, the failure to show that H3S28p is modulated at interphase heterochromatin foci is also a key missing element. Because an effect of RepoMan on heterochromatin formation was already published by these authors (Dev Cell 2011), I think a convincing molecular model for the underlying mechanism is required for publication in Nat Comms and this, unfortunately, is not quite provided by the current work.

Respectfully, we do not completely agree with the referee's vision on this point. In this paper we have clearly shown that Repo-Man is not only important to maintain heterochromatin but also that it is sufficient to generate heterochromatin in new chromosomal regions. Together with this, we have for the first time shown that Repo-man levels can produce changes in gene expression and chromatin positioning, all aspects that were not known before and that provide to a wider scientific community the notions that this complex within belongs to a class of important regulators of the epigenome.

Reviewer #4 (Remarks to the Author):

PRIDE is finally accessible and I have checked for the presence or the proper files on their website for this manuscript. The files are present and accessible, so the documentation of the proteomics data is in agreement with community standards.

We thank the referees for acknowledging this.